# IMBEDDING DEEP NEURAL NETWORKS

**Andrew Corbett**[*]
University of Exeter
a.j.corbett@exeter.ac.uk

**Dmitry Kangin**[*]
Etcembly Ltd.
dkangin@gmail.com

## ABSTRACT

Continuous-depth neural networks, such as Neural ODEs, have refashioned the understanding of residual neural networks in terms of non-linear vector-valued optimal control problems. The common solution is to use the adjoint sensitivity method to replicate a forward-backward pass optimisation problem. We propose a new approach which explicates the network's 'depth' as a fundamental variable, thus reducing the problem to a system of forward-facing initial value problems. This new method is based on the principle of 'Invariant Imbedding' for which we prove a general solution, applicable to all non-linear, vector-valued optimal control problems with both running and terminal loss. Our new architectures provide a tangible tool for inspecting the theoretical–and to a great extent unexplained–properties of network depth. They also constitute a resource of discrete implementations of Neural ODEs comparable to classes of imbedded residual neural networks. Through a series of experiments, we show the competitive performance of the proposed architectures for supervised learning and time series prediction. Accompanying code is made available at github.com/andrw3000/inimnet.

## 1 UNPACKING CONTINUOUS-DEPTH NEURAL NETWORKS

The long-standing enigma surrounding machine learning still remains paramount today: What is it that machines are learning and how may we extract meaningful knowledge from trained algorithms? Deep Neural Networks (DNNs), whilst undeniably successful, are notorious black-box secret keepers. To solve a supervised learning process, mapping vector inputs $\mathbf{x}$ to their targets $\mathbf{y}$, parameters are stored and updated in ever deepening layers with no facility to access the physical significance of the internal function approximating the global mapping $\mathbf{x} \mapsto \mathbf{y}$.

We propose a new class of DNNs obtained by *imbedding* multiple networks of varying depth whilst keeping the inputs, $\mathbf{x}$, *invariant*; we call these 'Invariant Imbedding Networks' (InImNets). To illustrate the concept, Figure 1 depicts a system of projectiles fired from a range of positions $p_1 < p_2 < \cdots < p_n$ with the same initial velocity conditions $\mathbf{x}$. The red curve (initiated at $p_1$) is fit to a sample (circles) along a single trajectory, representing a traditional regression problem. InImNet architectures are trained on the output values $\mathbf{y} = \mathbf{y}(p_i, \mathbf{x})$ at $p_n$ (the diamonds) as the depth $p_i$ of the system varies. This analogy applies to DNN classifiers where increasing the depth from $p_i$ to $p_{i-1}$ outputs a classification decision for each of the $i$-steps.

As a machine learning tool, the use of deep hidden layers, whilst successful, was first considered *ad hoc* in implementations, such as in multilayer

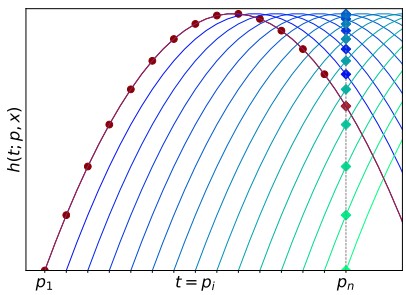

Figure 1: Plotted are heights $h(t; p, \mathbf{x})$ vs. lengths $t$ of projectile curves initiated with identical initial velocities $\mathbf{x}$ at a range of points $p_i$ along the $t$-axis. The red curve depicts a regression fit to a $t$-varying sample set; contrast with the blue InImNet training paradigm which learns the endpoints (diamonds) from varying the input position $p_i$.

---

[*]Equal Contribution.

perceptrons. But following the advent of residual neural networks (He et al., 2015) which use 'Euler-step' internal updates between layers, DNN evolution is seen to emulate a continuous dynamical system (Lu et al., 2018; Ruthotto & Haber, 2020). Thus was formed the notion of a 'Neural ODE' (Chen et al., 2018) in which the hidden network state vector $\mathbf{z}(t) \in \mathbb{R}^N$, instead of being defined at fixed layers $t \in \mathbb{N}$, is allowed to vary continuously over an interval $[p, q] \subset \mathbb{R}$ for a system of dimension $N \geq 1$. Its evolution is governed by an Ordinary Differential Equation (ODE)

$$\dot{\mathbf{z}}(t) = \mathbf{f}(t, \mathbf{z}(t), \boldsymbol{\theta}(t)); \quad \mathbf{z}(p) = \mathbf{x} \tag{1}$$

where the training function $\mathbf{f}$ is controlled by a parameter vector $\boldsymbol{\theta}(t) \in \mathbb{R}^M$ for $t \in [p, q]$ of length $M \geq 1$. Network outputs are retrieved at $\mathbf{z}(q) = \mathbf{y}$ after fixing the endpoint $t = q$. As such, the enigmatic 'depth' of a Neural ODE is controlled by varying $t = p$, at which point we insert the initial condition $\mathbf{z}(p) = \mathbf{x}$. This dynamical description has given the theory of DNNs a new home: the mathematical framework of optimal control (Massaroli et al., 2020; Bensoussan et al., 2020). In this framework, whether discrete or continuous, solution networks are sought after that: (A) satisfying their update law (1) over a fixed 'depth' interval $[p, q]$; (B) minimise a loss function subject to terminal success and internal regulation (see §2.1). As initiated by Euler and Lagrange (Euler, 1766), this mathematical framework determines networks given by (A) satisfying condition (B) using, amongst other techniques, *the adjoint method*: here a new state, which we call $\boldsymbol{\lambda}(t)$ or the 'Lagrange multiplier', is introduced containing the system losses with respect to both $t$ and the parameters $\boldsymbol{\theta}(t)$; see §2.3.

The connection between the traditional method of 'backpropagation-by-chain rule' and the rigorous 'adjoint method' is quite brilliantly explicated in proof by Chen et al. (2018), in that they directly deduce the adjoint method using infinitesimal backpropagation–far from the train of thought of Lagrange's multiplier method (Liberzon, 2012, Ch. 2); see also §D and §E.2 in the appendix for extended discussion and derivation.

Even within the above theory, the initial condition, $\mathbf{z}(p) = \mathbf{x}$, and location, $t = p$, remain implicit constraints; a clear understanding of network depth remains illusive.

Our new class of InImNet architectures may be obtained by *imbedding* networks of varying depth $p$ whilst keeping the inputs, $\mathbf{x}$, *invariant*. Explicating these two variables throughout the network, writing $\mathbf{z}(t) = \mathbf{z}(t; p, \mathbf{x})$, has exciting conceptual consequences:

1. **Forward pass to construct multiple networks:** InImNet state vectors $\mathbf{z}(t; p, \mathbf{x})$ are computed with respect to the depth variable $p$ rather than $t \in [p, q]$, which is considered fixed (in practice at $t = q$). We build from the bottom up: initiate at $p = q$ with the trivial network $\mathbf{z}(q; q, \mathbf{x}) = \mathbf{x}$ and unwind the $p$-varying dynamics, as described in Theorem 1, by integrating

   $$\nabla_p \mathbf{z}(q; p, \mathbf{x}) = -\nabla_{\mathbf{x}} \mathbf{z}(q; p, \mathbf{x}) \cdot \mathbf{f}(p, \mathbf{x}, \boldsymbol{\theta}(p)) \tag{2}$$

   from $p = q$ to a greater depth $p$. Note that at depth $p$ an InImNet returns an *external* output $\mathbf{z}(q; p, \mathbf{x}) \sim \mathbf{y}$, subject to training. This contrasts with convention, where one would obtain $\mathbf{z}(q; p, \mathbf{x})$ by integrating from $t = p$ to $t = q$, where $t < q$ states are considered *internal*. A general algorithm to implement the forward pass is described in Algorithm 1. The gradient operator $\nabla$ denotes the usual vector, or Jacobian, of partial derivatives.

2. **Backpropagate independently from the forward pass:** We generalise the adjoint method of Chen et al. (2018), who was able to do away with the backpropagation-by-chain rule method in favour of a continuous approach with at most bounded memory demand. With our bottom-up formulation, we are able to go one step further and do away with the initial forward pass altogether by initiating our 'imbedded' adjoint $\boldsymbol{\Lambda}(p, \mathbf{x})$, generalising $\boldsymbol{\lambda}(t)$, with loss gradients for the trivial network $\mathbf{z}(q; q, \mathbf{x}) = \mathbf{x}$ and computing to depth $p$ via

   $$\nabla_p \boldsymbol{\Lambda}(p, \mathbf{x}) = -[\nabla_{\mathbf{x}} \boldsymbol{\Lambda}(p, \mathbf{x}) \cdot \mathbf{f}(p, \mathbf{x}, \boldsymbol{\theta}(p)) + \nabla_{\mathbf{x}} \mathbf{f}(p, \mathbf{x}, \boldsymbol{\theta}(p))^{\mathrm{T}} \cdot \boldsymbol{\Lambda}(p, \mathbf{x})]. \tag{3}$$

   See Theorem 2 for a precise, more general explication. Backward passes may be made independently of forward passing altogether; see Theorem 2 and Algorithm 1.

3. **Pre-imposed optimality:** Working in the framework of optimal control theory, we consider both running and terminal losses–a general 'Bolza problem'–see §2.1. We give a necessary first-order criterion for optimal control (Theorem 3). In this way, we account for $t$-varying parameter controls $\boldsymbol{\theta}(t) = \boldsymbol{\theta}(t; p, \mathbf{x})$, omitted from the original Neural ODEs, permitting future compatibility with the recent particle-shooting models of Vialard et al. (2020).

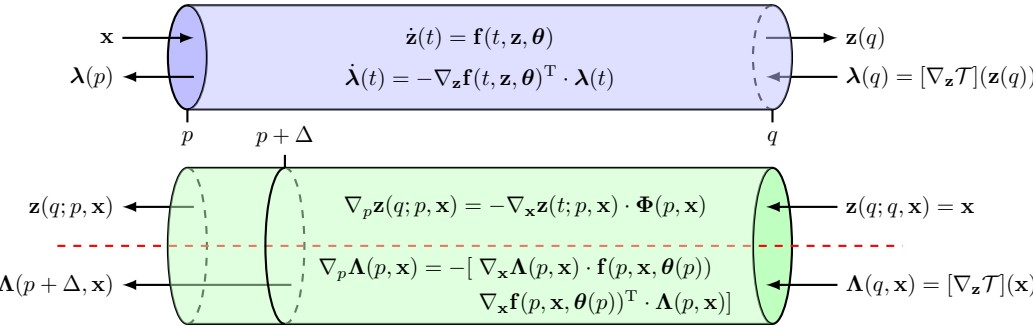

Figure 2: **Top:** A Neural ODE constitutes a two-point boundary value problem over $t \in [p, q]$. **Bottom:** An InImNet separates the forward and backward passes into separate initial value problems along the depth variable $p$.

**Our contribution.**   We prove that it is possible to reduce the non-linear, vector-valued optimal control problem to a system of forward-facing initial value problems. We introduce a new theoretical leap for the understanding of depth in neural networks, in particular with respect to viewing DNNs as dynamical systems (Chen et al., 2018; Massaroli et al., 2020; Vialard et al., 2020). We demonstrate that this approach may be used in creating discrete and continuous numerical DNN schemes. We verify such algorithms by successfully applying our method to benchmark experiments, which perform well on a range of complex tasks (high-dimensional rotating MNIST and bouncing balls) in comparison to state-of-the-art techniques.

**Architecture.**   In §3 we give novel 'InImNet' architectures implementing the above results. These may be applied to regression and classification problems using discrete or continuous algorithms.

**Experimental performance.**   In §4 we present some promising results for their practical functionality. These are designed to support the theoretical development of InImNets in §2 & 3 and support the proof of concept in application. We derive various successful discrete implementations, based on computing the state evolutions (2) and (3) with an Euler-step method. We also identify operational aspects in InImNets which we expect to encourage ongoing research and development. Crucially, our architectures are compatible with the various technical advancements since made to Neural ODEs, including those by Dupont et al. (2019); Davis et al. (2020); Yıldız et al. (2019); see also a study of effective ODE solvers for the continuous models (Hopkins & Furber, 2015).

**Broader impact for optimal control theory.**   Further afield, our result applies more generally as a new tool in the theory of optimal control. The mathematical technique that we apply to (1), deriving (2) and (3), is known as the Invariant Imbedding Method. The key output of the method is the reformulation of a two-point boundary value problem as a system of initial value problems, given as functions of initial value $\mathbf{x}$ and input location $p$ alone (Meyer, 1973). This stands in the literature as an alternative to applying the calculus of variations (Liberzon, 2012; Vialard et al., 2020). For linear systems, the technique was first developed by Ambarzumian (1943) to study deep stellar atmospheres. It has since found widespread applications in engineering (Bellman & Wing, 1975), optical oceanography (Mobley, 1994), PDEs (Maynard & Scott, 1971), ODEs (Agarwal & Saraf, 1979) and control theory (Bellman et al., 1966; Kalaba & Sridhar, 1969), to name a few. The non-linear case is only touched on in the literature for scalar systems of zero terminal loss (Bellman et al., 1966; Kalaba & Sridhar, 1969)–including some numerical computations to support its efficiency (Spingarn, 1972). In this work we derive a complete invariant imbedding solution, fit for applications such as those mentioned above, for a non-linear, vector-valued Bolza problem.

**Overview of the paper.**   In §2 we give the main theorems used for InImNet architectures and more widely in the field of optimal control. Detailed derivations and proofs may be found in the appendix, §E. In §3 we put forward various architectures to illustrate how InImNets may be utilised in learning paradigms. In §4 we describe our supporting experimental work for such architectures.

## 2 OPTIMISATION VIA INVARIANT IMBEDDING

Solutions to (1) depend *implicitly* on both an input datum $\mathbf{x}$ and the input position $p$ at which the input is cast. The $(p, \mathbf{x})$ relationship is at the heart of the invariant imbedding method which *explicates* these arguments, written into the notation as $\mathbf{z}(t) = \mathbf{z}(t; p, \mathbf{x})$. In this section we begin by introducing the optimisation problem before giving our invariant imbedding solution. Fix integers $M, N \geq 1$ to denote the dimension of the instantaneous parameter space $\boldsymbol{\theta}(t) \in \mathbb{R}^M$ and the dynamical system $\mathbf{z}(t), \mathbf{f}(t, \mathbf{z}, \boldsymbol{\theta}) \in \mathbb{R}^N$.

### 2.1 THE BOLZA OPTIMISATION PROBLEM

The key advantage of studying continuous DNNs is to redress the learning problem as an optimal control problem. Here we consider the most general control problem, a *Bolza problem*, which subject to two forms of loss: a *terminal loss*, the discrepancy between the system outputs $\mathbf{z}(q)$ and the true outputs $\mathbf{y}$ measured by a loss function $\mathcal{T}$ on $\mathbb{R}^N$; and a *running loss*, which regulates both the state vector $\mathbf{z}(t)$ itself as it evolves over $t \in [p, q]$ but also the control $\boldsymbol{\theta}(t)$ over $[p, q]$ by a square-integrable functional $\mathcal{R}$ on $[p, q] \times \mathbb{R}^N \times \mathbb{R}^M$. Together, the minimisation problem is to find a control $\boldsymbol{\theta}(t)$ and solution $\mathbf{z}(t)$ to (1) whilst minimising the total loss

$$\mathcal{J}(\boldsymbol{\theta}; p, \mathbf{x}) := \int_p^q \mathcal{R}(t, \mathbf{z}(t; p, \mathbf{x}), \boldsymbol{\theta}(t; p, \mathbf{x}))dt + \mathcal{T}(\mathbf{z}(q; p, \mathbf{x})). \tag{4}$$

for each known datum pair $(\mathbf{x}, \mathbf{y})$. Applying the calculus of variations, one considers small perturbations about an optimal control $\boldsymbol{\theta}$ which minimise $\mathcal{J}(\boldsymbol{\theta}; p, \mathbf{x})$ whilst determining a solution $\mathbf{z}$ to (1). The well known first-order Euler–Lagrange optimality equations (see §E.3) thus derived constitute a constrained, two-point boundary value problem as depicted in Figure 2. By contrast, the invariant imbedding method restructures the first-order optimal system as an initial value problem. The $t$-dependence is brushed implicitly under the rug, with numerical integration performed instead on the depth variable $p$.

### 2.2 THE INVARIANT IMBEDDING SOLUTION

The fundamental principle we follow is to observe the imbedding of intervals $[p + \Delta, q] \subset [p, q]$, for $0 \leq \Delta \leq p - q$, which carry solutions $\mathbf{z}(t; p + \Delta, \mathbf{x})$ to (1), whilst keeping the input, $\mathbf{x}$, to each invariant. In limiting terms as $\Delta \to 0$, the partial rate of change in depth $\nabla_p = \partial/\partial p$ is directly related to the vector gradient $\nabla_{\mathbf{x}}$ for the initial value $\mathbf{x}$ at $p$. This is controlled by the coefficient

$$\boldsymbol{\Phi}(p, \mathbf{x}) := \mathbf{f}(p, \mathbf{x}, \boldsymbol{\theta}(p)). \tag{5}$$

**Theorem 1.** *Let $\boldsymbol{\theta}(t)$ be an admissible control, by which we mean $\boldsymbol{\theta}$ is piecewise continuous on $t \in [p, q]$. Suppose the dynamics function $\mathbf{f}: [p, q] \times \mathbb{R}^N \times \mathbb{R}^M \to \mathbb{R}^N$ is continuous on $(t, \boldsymbol{\theta}) \in [p, q] \times \mathbb{R}^M$ and continuously differentiable on $\mathbf{z} \in \mathbb{R}^N$. Let $t \in [p, q]$ and suppose $\mathbf{z}(t; p, \mathbf{x})$ and $\boldsymbol{\theta}(t; p, \mathbf{x})$ satisfy (1) for each $\mathbf{x} \in \mathbb{R}^N$. Then we have the invariant imbedding relation*

$$\nabla_p \mathbf{z}(t; p, \mathbf{x}) = -\nabla_{\mathbf{x}} \mathbf{z}(t; p, \mathbf{x}) \cdot \boldsymbol{\Phi}(p, \mathbf{x}). \tag{6}$$

The assumptions in Theorem 1 could be relaxed further. For instance, one could expand the class of admissible controls $\boldsymbol{\theta}(t)$ to those which are measurable and locally bounded. The dynamics function could relax the existence assumption on $\nabla_{\mathbf{z}}\mathbf{f}$ in replace of a Lipschitz property. The assumptions stated are permissible for a wide range of applications. One may read more about possible weaker hypotheses in (Liberzon, 2012, §3.3.1).

We use this result given by (6) as a model to address the following learning problem. Consider a collection of input values $\mathbf{x} \in \mathbb{R}^N$ corresponding to known output values $\mathbf{y} \in \mathbb{R}^N$. We seek to extend an approximation of $\mathbf{x} \mapsto \mathbf{y}$ to larger subsets of $\mathbb{R}^N$. We proceed by choosing an interval $[p, q] \subset \mathbb{R}$ of arbitrary depth (fixing $q$ and varying $p$) and postulate a state vector $\mathbf{z}(t; p, \mathbf{x})$, subject to (1), that approximates $\mathbf{y}$ at $t = q$ given the input $\mathbf{z}(p; p, \mathbf{x}) = \mathbf{x}$.

The parameter control, whilst commonly restricted in applications, is *a priori* subject to the same dependencies $\boldsymbol{\theta}(t) = \boldsymbol{\theta}(t; p, \mathbf{x})$. We denote a second coefficient related to its endpoint by

$$\boldsymbol{\Psi}(p, \mathbf{x}) := \boldsymbol{\theta}(p; p, \mathbf{x}) \tag{7}$$

which, for $p \leq t \leq q$, also satisfies the invariant imbedding relation in Theorem 1:

$$\nabla_p \boldsymbol{\theta}(t; p, \mathbf{x}) = -\nabla_{\mathbf{x}} \boldsymbol{\theta}(t; p, \mathbf{x}) \cdot \boldsymbol{\Phi}(p, \mathbf{x}). \tag{8}$$

Whilst this observation may be made of the underlying dynamical system alone, the consequences extend to the control problem in §2.1 and form the basis of our solution.

### 2.3 BACKWARD LOSS PROPAGATION

The calculus of variations, introduced by Euler and Lagrange (Euler, 1766), provides a mechanism by which to find an *optimal control* $\boldsymbol{\theta}$ (see Liberzon, 2012, Ch. 2). The key trick is to invoke a function called the Lagrange multiplier $\boldsymbol{\lambda}(t) = \boldsymbol{\lambda}(t; p, \mathbf{x})$, also known as the "adjoint state" (Chen et al., 2018) or "Hamiltonian momentum" (Vialard et al., 2020), which encodes the backward-propagated losses. Indeed, $\boldsymbol{\lambda}(t; p, \mathbf{x})$, the initial value at the endpoint $t = q$, is defined to be the gradient of the terminal loss $\mathcal{T}(\mathbf{z}(q; p, \mathbf{x}))$ with respect to $\mathbf{z}(q; p, \mathbf{x})$. To optimise the network, whether directly using (12) or by stochastic gradient descent on the weights, one must propagate $\boldsymbol{\lambda}(t; p, \mathbf{x})$ back to $t = p$. To this end we introduce the 'imbedded adjoint' coefficient

$$\boldsymbol{\Lambda}(p, \mathbf{x}) = \boldsymbol{\lambda}(p; p, \mathbf{x}) \tag{9}$$

which is the subject of our main backward result.

**Theorem 2.** *Suppose that $\boldsymbol{\theta}$, $\mathbf{z}$ and $\mathbf{f}$ satisfy the hypotheses of Theorem 1 for each $p \leq q$. Suppose too that the terminal loss, $\mathcal{T}$, and running loss, $\mathcal{R}$, are defined as in Equation (4), subject to the hypotheses of §2.1. Then the imbedded adjoint $\boldsymbol{\Lambda}(p, \mathbf{x})$ satisfies the reverse initial value problem*

$$-\nabla_p \boldsymbol{\Lambda}(p, \mathbf{x}) = \nabla_{\mathbf{x}} \boldsymbol{\Lambda}(p, \mathbf{x}) \cdot \boldsymbol{\Phi}(p, \mathbf{x}) + [\nabla_{\mathbf{z}} \mathbf{f}](p, \mathbf{x}, \boldsymbol{\Psi}(p, \mathbf{x}))^{\mathrm{T}} \cdot \boldsymbol{\Lambda}(p, \mathbf{x}) + [\nabla_{\mathbf{z}} \mathcal{R}](p, \mathbf{x}, \boldsymbol{\Psi}(p, \mathbf{x})) \tag{10}$$

*initiated at $p = q$ by the value $\boldsymbol{\Lambda}(q, \mathbf{x}) = [\nabla_{\mathbf{z}} \mathcal{T}](\mathbf{x})$.*

The bracketed use of $[\nabla_{\mathbf{z}} \mathbf{f}]$ etc. is to stress that the differential operator acts before evaluation.

We contrast our approach of fixed $t = q$ and varying depth $p$ with the adjoint method of Chen et al. (2018), who fix $p$ and vary $p \leq t \leq q$. Our derivation provides a new proof of the standard Euler–Lagrange equations which give the adjoint method, manifesting in our account as

$$\boldsymbol{\lambda}(p; p, \mathbf{x}) = [\nabla_{\mathbf{z}} \mathcal{T}](\mathbf{z}(q; p, \mathbf{x})) - \int_q^p ([\nabla_{\mathbf{z}} \mathbf{f}](t, \mathbf{z}, \boldsymbol{\theta})^{\mathrm{T}} \cdot \boldsymbol{\lambda}(t; p, \mathbf{x}) + [\nabla_{\mathbf{z}} \mathcal{R}](t, \mathbf{z}, \boldsymbol{\theta})) dt \tag{11}$$

with initial value $\boldsymbol{\lambda}(q; p, \mathbf{x}) = [\nabla_{\mathbf{z}} \mathcal{T}](\mathbf{z}(q; p, \mathbf{x}))$ at $t = p$. Observe that in Theorem 2 the initial loss at $p = q$ is given by $[\nabla_{\mathbf{z}} \mathcal{T}](\mathbf{z}(q; q, \mathbf{x})) = [\nabla_{\mathbf{z}} \mathcal{T}](\mathbf{x})$, for the trivial network. Back-integrating this term thus does not require a forward pass of the $\mathbf{z}$-state at the cost of computing the derivatives with respect to $\nabla_{\mathbf{x}} \boldsymbol{\Lambda}(p, \mathbf{x})$. Optimising this latter process opens a new window of efficient DNNs.

### 2.4 A FIRST ORDER OPTIMALITY CONDITION

With the insights gleaned from optimal control theory, Vialard et al. (2020) take a different approach facilitating $t$-varying parameter controls $\boldsymbol{\theta}(t; p, \mathbf{x})$. This is based on assuming that $\boldsymbol{\theta}$ is optimal from the outset. This is achieved through specifying $\boldsymbol{\theta}$ by the $t$-varying constraint

$$[\nabla_{\boldsymbol{\theta}} \mathcal{R}](t, \mathbf{z}, \boldsymbol{\theta}) + [\nabla_{\boldsymbol{\theta}} \mathbf{f}](t, \mathbf{z}, \boldsymbol{\theta})^{\mathrm{T}} \cdot \boldsymbol{\lambda}(t; p, \mathbf{x}) = 0. \tag{12}$$

We obtain a corresponding condition for the coefficients that constitute an InImNet.

**Theorem 3.** *Suppose that $\boldsymbol{\theta}$, $\mathbf{z}$ and $\mathbf{f}$ satisfy the hypotheses of Theorem 1 for each $p \leq q$. Suppose the losses $\mathcal{T}$, $\mathcal{R}$ and $\mathcal{J}$ are defined as in (4) subject to the hypotheses of §2.1. Then the first-order optimality condition for the total loss $\mathcal{J}(\boldsymbol{\theta}; p, \mathbf{x})$ to be minimised is given by*

$$[\nabla_{\boldsymbol{\theta}} \mathcal{R}](p, \mathbf{x}, \boldsymbol{\Psi}(p, \mathbf{x})) + [\nabla_{\boldsymbol{\theta}} \mathbf{f}](p, \mathbf{x}, \boldsymbol{\Psi}(p, \mathbf{x}))^{\mathrm{T}} \cdot \boldsymbol{\Lambda}(p, \mathbf{x}) = 0. \tag{13}$$

Making the $(p, \mathbf{x})$-dependency explicit for optimal controls, this identity provides a mechanism by which the depth $p$ itself is accessible to optimise. In practice, $\boldsymbol{\Psi}(p, \mathbf{x})$, and hence $\boldsymbol{\Phi}(p, \mathbf{x})$, are derived from $\boldsymbol{\Lambda}(p, \mathbf{x})$; these are connected by Theorem 3 above. Altogether, Equations (6), (8), (10) and (13) constitute the invariant imbedding solution to the general Bolza problem as advertised.

## 3 InImNet Architectures

The architectures presented here are based upon the results of Theorems 1, 2 & 3. Whilst the processes for obtaining the $p$-th network state $\mathbf{z}(q; p, \mathbf{x})$ and backward adjoint $\mathbf{\Lambda}(p, \mathbf{x})$ are independent processes, we nevertheless describe their general form together in Algorithm 1.

---

**Algorithm 1** Independent forward and backward pass with InImNet

---

**Require:** Training set of input/output pairs $(\mathbf{x}, \mathbf{y})$; evaluation points $p_1 < \cdots < p_n$; loss function $\mathcal{T}$ (see §2.1); training function $\mathbf{f}(t, \mathbf{z}, \boldsymbol{\theta})$.

**Ensure:** Track $\mathbf{x}$-operations for auto-differentiation, or substitute a numerical derivative (see §3.2).

 **Inputs:** $\mathbf{z}(q; p_i, \mathbf{x}) = \mathbf{x}$; $\mathbf{\Lambda}(q, \mathbf{x}) = \nabla_{\mathbf{x}} \mathcal{T}(\mathbf{x})$

 **for** $i = n - 1, \ldots, 1$ **do**

  $\mathbf{z}(q; p_i, \mathbf{x}) = \mathbf{z}(q; p_{i+1}, \mathbf{x}) + \int_{p_{i+1}}^{p_i} \nabla_p \mathbf{z}(q; p, \mathbf{x})$       ▷ Use Theorem 1.

  $\mathbf{\Lambda}(p_i, \mathbf{x}) = \mathbf{\Lambda}(p_{i+1}, \mathbf{x}) + \int_{p_{i+1}}^{p_i} \nabla_p \mathbf{\Lambda}(p, \mathbf{x})$       ▷ Use Theorem 2.

 **end for**

 **Returns:** Tuple of outputs $\mathbf{z}(q; p_i, \mathbf{x})$ corresponding to networks of varying depths $p_i$.

---

In the remainder of this section we describe various discrete models to implement variants of Algorithm 1. Continuous architectures using black-box auto-differentiable ODE solvers, such as those considered by Chen et al. (2018), may be readily implemented. This approach poses interesting new avenues of research based on implementing accurate numerical alternatives to the computation of nested Jacobians. Simultaneously, the question of stability of DNN dynamics functions becomes another crucial question to address.

For our experiments we seek to show a first-principle implementation of InImNets, and we do so by describing a discrete architecture, executing the minimum computation time whilst demonstrating high performance; see §3.1.

Finally, time-series data, or running losses, are not considered by Algorithm 1 but may be solved by InImNet, their dynamical structure a natural formulation for InImNets. We consider such an architecture in §C of the appendix as well as its application to regression problems in §F.2.

### 3.1 Euler-method experimental architecture

For implementation, we pass through the underlying ODEs with a proposed architecture based on a simple forward-Euler solution to the integrals in Algorithm 1. This is comparable to the original ResNet architectures (He et al., 2015). To do this we divide up the interval into a collection of layers $[p, q] = \cup_{i=1}^{n-1} [p_i, p_{i+1}]$ and rewrite the invariant imbedding equation of Theorem 1 as

$$\mathbf{z}(t; p_i, \mathbf{x}) = \mathbf{z}(t; p_{i+1}, \mathbf{x}) - (p_i - p_{i+1}) \nabla_{\mathbf{x}} \mathbf{z}(t; p_{i+1}, \mathbf{x}) \cdot \mathbf{\Phi}(p_i, \mathbf{x}), \tag{14}$$

subject to $\mathbf{z}(p_n; p_n, \mathbf{x}) = \mathbf{x}$. Backpropagation may then be executed by either differentiating through the system, as is the standard approach, or by implementing Theorem 2 through the forward-Euler formula

$$\mathbf{\Lambda}(p_i, \mathbf{x}) = \mathbf{\Lambda}(p_{i+1}, \mathbf{x})$$
$$- (p_i - p_{i+1})[\nabla_{\mathbf{x}} \mathbf{\Lambda}(p_i, \mathbf{x}) \cdot \mathbf{\Phi}(p_i, \mathbf{x})) + \nabla_{\mathbf{x}} \mathbf{f}(p_i, \mathbf{x}, \boldsymbol{\theta}(p_i))^{\mathrm{T}} \cdot \mathbf{\Lambda}(p_{i+1}, \mathbf{x})] \tag{15}$$

with the initial condition $\mathbf{\Lambda}(p_n, \mathbf{x}) = \nabla_{\mathbf{x}} \mathcal{T}(\mathbf{x})$. For our experimental applications, we also apply a technique to approximate the inherent Jacobians within the InImNet architecture; see Equation (21) in §3.2.

### 3.2 Numerical approximation of input gradients

An implicit computational speed bump is the computation of the gradients $\nabla_{\mathbf{x}}$ in (2), (6) and (10). The immediate solution is to track the gradient graphs of these terms with respect to $\mathbf{x}$ and implement automatic differentiation. Indeed, this approach does yield successful models–if one has time on their hands but the drawback is that a high memory cost is incurred for deep or high dimensional networks.

We offer some surrogate numerical solutions. For the sake of example, suppose we wish to compute $\mathbf{z}(q; p, \mathbf{x}) \in \mathbb{R}^N$ by integrating

$$\nabla_p \mathbf{z}(q; p, \mathbf{x}) = -\nabla_\mathbf{x} \mathbf{z}(q; p, \mathbf{x}) \cdot \mathbf{\Phi}(p, \mathbf{x}) \tag{16}$$

with respect to $p$. To compute the derivatives $\nabla_\mathbf{x}$ we consider perturbations of the input vector $\mathbf{x} \in \mathbb{R}^N$ of the form $\mathbf{x} \pm \Delta_i \mathbf{e}_i$ for appropriately small $\Delta_i > 0$ and $\mathbf{e}_i := (\delta_{ij})_{1 \leq j \leq N}$ for $i = 1, \ldots, N$. We then solve for the $2N + 1$ states $\mathbf{z}(q; p, \mathbf{x} \pm \Delta_i \mathbf{e}_i)$ by simultaneously integrating (16) alongside

$$\nabla_p \mathbf{z}(q; p, \mathbf{x} \pm \Delta_i \mathbf{e}_i) = -\nabla_\mathbf{x} \mathbf{z}(q; p, \mathbf{x} \pm \Delta_i \mathbf{e}_i) \cdot \mathbf{\Phi}(p, \mathbf{x} \pm \Delta_i \mathbf{e}_i) \tag{17}$$

where the gradients $\nabla_\mathbf{x} \mathbf{z}(q; p, \mathbf{x}_0)$ are modelled by

$$\nabla_\mathbf{x} \mathbf{z}(q; p, \mathbf{x}_0) \approx \left[ \frac{\mathbf{z}(q; p, \mathbf{x}_0 + \Delta_i \mathbf{e}_i) - \mathbf{z}(q; p, \mathbf{x}_0 - \Delta_i \mathbf{e}_i)}{2\Delta_i} \right]_i \tag{18}$$

for $\mathbf{x}_0 = \mathbf{x}, \mathbf{x} \pm \Delta_i \mathbf{e}_i$, respectively. This method is known as the symmetric difference quotient. Other approximations may also be applied on a bespoke basis, such as Newton's difference quotient. This uses a similar construction but the negative shifts are forgotten, resulting in tracking $N + 1$ equations along (16) and (17) where we estimate

$$\nabla_\mathbf{x} \mathbf{z}(q; p, \mathbf{x}) \approx \left[ \frac{\mathbf{z}(q; p, \mathbf{x} + \Delta_i \mathbf{e}_i) - \mathbf{z}(q; p, \mathbf{x})}{\Delta_i} \right]_i \tag{19}$$

and

$$\nabla_\mathbf{x} \mathbf{z}(q; p, \mathbf{x} + \Delta_i \mathbf{e}_i) \approx -\nabla_\mathbf{x} \mathbf{z}(q; p, \mathbf{x}). \tag{20}$$

Alternatively, and more directly, we may tackle computing the successive Jacobians $\nabla_\mathbf{x} \mathbf{z}(t, p, \mathbf{x})$, incurring a high memory cost storing gradient graphs, by approximating such terms through cropping the higher order gradients:

$$\nabla_\mathbf{x} \mathbf{z}(t; p_i, \mathbf{x}) = \nabla_\mathbf{x} \mathbf{z}(t; p_{i+1}, \mathbf{x}) - \nabla_\mathbf{x} \nabla_\mathbf{x} \mathbf{z}(t; p_{i+1}, \mathbf{x}) \cdot \mathbf{\Phi}(p_i, \mathbf{x})$$
$$\approx \nabla_\mathbf{x} \mathbf{z}(t; p_{i+1}, \mathbf{x}) - \nabla_\mathbf{x} \mathbf{\Phi}(p_i, \mathbf{x}) \tag{21}$$

Whilst theoretically losses are easily quantifiable, we show experimentally that for this approximation an increasing the number of layers still improves the performance of the model.

## 4 EXPERIMENTAL RESULTS

In this section we demonstrate the practical ability of the proposed architectures in solving benchmark problems for deep neural networks. All experiments use backpropagation to differentiate through the system as outlined in §3.1 except for the projectile motion study in the appendix, §F.2, where training is executed via the update rule stated by Equation (15). Without loss of generality, we arbitrarily choose $q = 0$ and assume $p$ to be in the range $[p_{\min}, 0]$ where $p_{\min}$ is a negative value whose modulus represents 'depth' of the network.

### 4.1 BENCHMARK VALIDATION

#### 4.1.1 ROTATING MNIST

We replicate the experimental setting used by Yıldız et al. (2019) and Vialard et al. (2020). This gives a point of comparison between our proposed InImNet, for various depth architectures, as described in §3.1.

In the 'Rotating MNIST' experiment, we solve the task of learning to generate the handwritten digit '3' for a sequence of sixteen equally spaced rotation angles between $[0, 2\pi]$ given only the first example of the sequence. To match the experimental setting with Yıldız et al. (2019) and Vialard et al. (2020), we train the model on layer $p_{\min}$ using backpropagation by minimising the objective of mean squared error for all angles, except for one fixed angle (in the fifth frame) as well as three random angles for each sample. We report the Mean Squared Error (MSE) at the fifth frame and the standard deviation over ten different random initialisations.

The results of this experiment are given in Table 1. The details of the experimental set-up and the hyperparameters are given in §G.1 of the appendix. The results show comparable performance to that achieved by (Vialard et al., 2020) while using a more computationally efficient discrete invariant imbedding formulation (see §4.1.2). We implement the InImNet dynamics function $\mathbf{\Phi}$ (as in Theorem 1) as a Multilayer Perceptron (MLP) with either two or three layers.

| Existing Models | | InImNet | | | |
| Method | MSE $\pm\sigma$ | $p_{\min}$ | $n$ | MSE $\pm\sigma$ | Time [s/epoch] |
|---|---|---|---|---|---|
| GPPVAE-DIS | $0.0309 \pm 0.00002$ | $-1$ | 2 | $0.0156 \pm 0.0008$ | 3.3459 |
| GPPVAE-JOINT | $0.0288 \pm 0.00005$ | $-2$ | 2 | $0.0130 \pm 0.0005$ | 4.1624 |
| ODE$^2$VAE | $0.0194 \pm 0.00006$ | $-3$ | 2 | $0.0126 \pm 0.0007$ | 5.0060 |
| ODE$^2$VAE-KL | $0.0184 \pm 0.0003$ | $-4$ | 2 | $0.0125 \pm 0.0004$ | 5.5806 |
| Vialard et al. (2020)* | $0.0122 \pm 0.0064$** | $-1$ | 3 | $0.0176 \pm 0.0010$ | 3.1504 |
| (*) 8.6363s/epoch | | $-2$ | 3 | $0.0129 \pm 0.0008$ | 4.0412 |
| | | $-3$ | 3 | $0.0125 \pm 0.0003$ | 4.886 |
| | | $-4$ | 3 | $0.0126 \pm 0.0004$ | 5.8521 |

Table 1: Rotating MNIST: Reported MSE for the proposed InImNet (where $n$ is the number of MLP layers) and state-of-the-art methods, results from other methods are taken from Yıldız et al. (2019) and Vialard et al. (2020). (**) The standard deviation given is more than half the mean value as stated in Vialard et al. (2020)

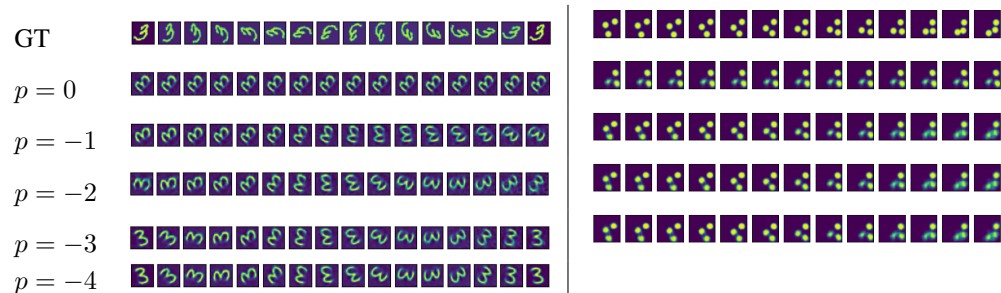

Figure 3: Left: samples from 'Rotating MNIST' experiment with $p_{\min} = -4$ and a two-layer MLP. Right: samples from 'Bouncing Balls' experiment with $p_{\min} = -3$ and a three-layer MLP. See §G.1 and §G.2 of the appendix for architectural details.

#### 4.1.2 BOUNCING BALLS

As in the previous section, we replicate the experimental setting of Yıldız et al. (2019) and Vialard et al. (2020). We use the experimental architecture given in §3.1 and we list hyperparameters used §G.2 of the appendix.

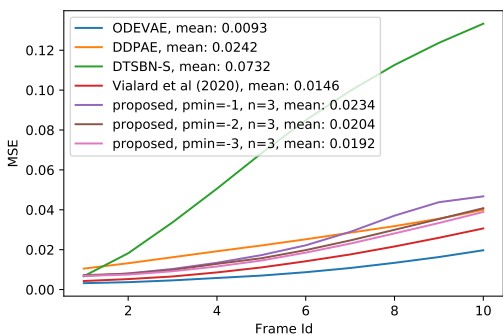

| Model | $p_{\min}$ | MSE | Time [s/epoch] |
|---|---|---|---|
| InImNet | $-1$ | 0.0234 | 91 |
| InImNet | $-2$ | 0.0204 | 121 |
| InImNet | $-3$ | 0.0192 | 153 |
| Vialard | | 0.0146 | 516 |

Figure 4: Bouncing balls experiment. Left: reported MSE for the proposed InImNet and state-of-the-art methods, results from other methods are taken from Yıldız et al. (2019) and Vialard et al. (2020). Right: average time consumption per epoch.

For this 'Bouncing Balls' experiment we note that the MSE of the proposed InImNet and the state-of-the-art methods are comparable while using a more computationally efficient model. We measured the time per epoch whilst using a public configuration of Google Colab for the (best-performing) up-down method of Vialard et al. (2020) against InImNet (with $p_{\min} = -3$; three-layer MLP). We set the batch size to the same value of 25 as given in the configuration of the official implementation in Vialard et al. (2020). While the proposed InImNet requires 153 seconds per epoch, the method as described by Vialard et al. (2020) took 516 seconds to finish one epoch.

## 4.2 EXTRAPOLATION BEYOND OPTIMISED MODELS

The key insight provided by InImNets is the simulation of multiple networks (for each $p$) from a single network pass. We give an expanded account of performance of this nature in §F.3 of the appendix, alongside the experimental set-up. To demonstrate such observations, we exhibit a prototype loss-plot for the Rotating MNIST experiment (in the appendix, see Figure 7). The horizontal axis varies with the frame of the rotation–the dip in loss corresponds to a neutral frame, as expected. Each line, $p = -1, -2, \ldots$, corresponds to an output at a different $p$-layer. The network is tuned (or trained) on its outputs at layer $p = p_{\min} = -4$. However, by extrapolating either side of $p_{\min}$ we can make quantitative observations on the rate of loss degradation; for example, the total loss remains lower shifting to $p = -5$ rather than $p = -3$.

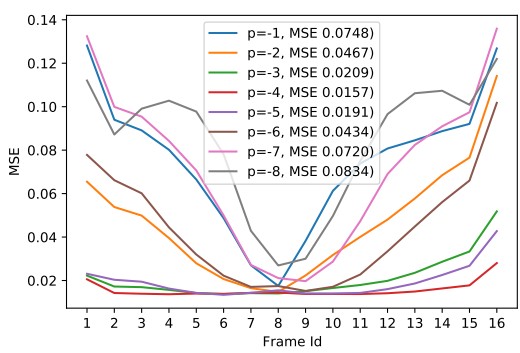

Figure 5: Extrapolation beyond and before chosen training depth at $p_{\min} = -4$.

## 5 DISCUSSION AND CONCLUSIONS

We have shown that it is possible to reduce the non-linear, vector-valued optimal control problem to a system of forward-facing initial value problems. We have demonstrated that this approach may be used in creating discrete and continuous numerical schemes. In our experiments, we show that (1) for a range of complex tasks (high-dimensional rotating MNIST and bouncing balls), the discrete model exhibits promising results, competitive with the current state-of-the-art methods while being more computationally efficient; and (2) the continuous model, via the Euler method, shows promising results on a projectile motion task. They demonstrate the potential for inference in continuous neural networks by using the invariant imbedding method to vary the initial conditions of the network rather than the well-known forward-backward pass optimisation.

We have outlined a class of DNNs which provide a new conceptual leap within the understanding of DNNs as dynamical systems. In particular, the explication of the depth variable leads to a new handle for the assessment of stacking multiple layers in DNNs. This also fits within the framework of Explainable AI, whereby an InImNet model is able to depict a valid output at every model layer.

Of course, nothing comes for free. The expense we incur is the presence of nested Jacobian terms; for example, $\nabla_{\mathbf{x}}\mathbf{z}(t; p, \mathbf{x})$. We show experimentally that our models perform well with elementary approximations for the purpose of functionality. But understanding these terms truly is deeply related to the stability of Neural ODEs over a training cycle.

In this article we do not explore the ramifications of the optimality condition of Theorem 3. With the work of (Vialard et al., 2020), in which systems are considered optimal from the outset via Theorem 3, we propose to study the variability of depth of such optimal systems.

We end where we started with the image of Figure 1. In the appendix, in §C and §F.2, we implement a time-series architecture and apply this to modelling projectile motion curves. We discuss the difficulties faced by InImNets for high-depth data in §F.1 and suggest a promising avenue for extended research.

ACKNOWLEDGMENTS

The first author would like to extend sincere thanks to Prof. Jacq Christmas and Prof. François-Xavier Vialard for their engagement on this topic.

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

## A   OVERVIEW OF NOTATION AND CONVENTIONS

We consider all $\mathbb{R}$-vectors as column vectors, to be denoted in bold font. Let $\langle \mathbf{v}, \mathbf{w} \rangle$ denote the Euclidean product between two vectors $\mathbf{v}$ and $\mathbf{w}$. Matrix multiplication between $\mathbf{A}$ and $\mathbf{B}$ shall be denoted $\mathbf{A} \cdot \mathbf{B}$ and transposition by $\mathbf{A}^T$. For $\mathbf{v} = (v_i)_i$ introduce the gradient operator by the row vector $\nabla_{\mathbf{v}} = (\partial/\partial v_i)_i$ which operates on scalar fields as usual but also on vector fields $\mathbf{F}(\mathbf{v}) = (F_i(\mathbf{v}))_i$ to give the Jacobian matrix $\nabla_{\mathbf{v}} \mathbf{F} = (\partial F_i/\partial v_j)_{i,j}$. Then, if $\mathbf{v} = \mathbf{v}(\mathbf{w})$, the chain rule reads $\nabla_{\mathbf{w}} \mathbf{F}(\mathbf{v}(\mathbf{w})) = \nabla_{\mathbf{v}} \mathbf{F} \cdot \nabla_{\mathbf{w}} \mathbf{v}$, which could be expressed as $\partial \phi / \partial w = \langle \nabla_{\mathbf{v}} \phi, \partial \mathbf{v}/\partial w \rangle$ if $\mathbf{F} = \phi$ and $\mathbf{w} = w$ were both scalar valued. Moreover we extend the gradient notation by using it for scalars $p \in \mathbb{R}$ to denote the single partial derivative $\nabla_p = \partial/\partial p$. Define the second order gradient operator by the matrix of second order partial derivatives $\nabla^2_{\mathbf{v},\mathbf{w}} := \nabla_{\mathbf{w}} \nabla_{\mathbf{v}} = (\partial^2/\partial v_i \partial w_j)_{i,j}$ with respect to $\mathbf{v} = (v_i)_i$ and $\mathbf{w} = (w_i)_i$. We use the Dirac delta symbol given by $\delta_{ij} = 1$ if and only if $i = j$ and $\delta_{ij} = 0$ otherwise. The $N \times N$ identity matrix is then given by $\mathbf{1}_N := (\delta_{ij})_{1 \le i,j \le N}$. Lastly, for a (possibly vector valued) function $\mathbf{F}$ on $\mathbb{R}$ we write $\mathbf{F}(x) = o(x)$, implicitly with respect to $x \to 0$, to denote that $\lim_{x \to 0} \mathbf{F}(x)/x = 0$.

## B   THE IMBEDDING RULE

InImNet architectures are based on the principle of invariant imbedding: the description of a system of a given length in terms of its imbedded sister systems. For example, consider two lengths $p_2 <$

$p_1 < q$ such that the solution $\mathbf{z}(t; p_1, \mathbf{x})$ to (1) is known for $t \geq p_1$. The solution $\mathbf{z}(t; p_2, \mathbf{x})$ over $[p_2, q]$ restricted to $t \geq p_1$, with $\mathbf{x}$ input at $p_2$, may be expressed using (6) by

$$\mathbf{z}(t; p_2, \mathbf{x}) = \mathbf{z}(t; p_1, \mathbf{x}) - \int_{p_1}^{p_2} \nabla_{\mathbf{x}} \mathbf{z}(t; p, y) \mathbf{\Phi}(p, \mathbf{x}) dp. \tag{22}$$

In this way, two separate intervals may be adjoined by observing the imbedding rule

$$\mathbf{z}(t; p_2, \mathbf{x}) = \mathbf{z}(t; p_1, \mathbf{z}(p_1; p_2, \mathbf{x})) \tag{23}$$

where the new input $\mathbf{z}(p_1; p_2, \mathbf{x})$ is evaluated using (22) at $t = p_1$. In the case that $\mathbf{f}$ is a linear function of $\mathbf{z}$, the identity in (22) is sometimes known as the Ricatti transformation (Bellman & Wing (1975)). In practical computations, one should devise models, such as step-linearisation, to solve (22) by simulating the gradients $\nabla_{\mathbf{x}} \mathbf{z}(p_1; p, \mathbf{x})$ for $p < p_2$.

## C    LEARNING LATENT DYNAMICS FROM END-POINT OBSERVATIONS

Here we propose an architechture for the application of InImNets to time-series regression problems, such as we implement in §F.2.

Typical regression problems ask for a model, given by a state $\mathbf{z}(t)$, to approximate a $t$-varying process $\mathbf{y}(t)$ sampled at a finite number of training points $\mathbf{y}(t_i)$ in a fixed region of interest $t_i \in [p, q]$. Assuming that $\mathbf{y}$ satisfies a well behaved ODE, the learning paradigm of a Neural ODE $\dot{\mathbf{z}} = \mathbf{f}$, see (1), is well suited to learn such continuous dynamics. Inherently, the initial condition $\mathbf{y}(p) = \mathbf{x}$ is assumed fixed.

InImNets are naturally structured to learn a variant of this problem: consider a $t$-varying process modelled by the dynamical system

$$\dot{\mathbf{y}}(t; p, \mathbf{x}) = \mathbf{g}(t, \mathbf{y}); \quad \mathbf{y}(p; p, \mathbf{x}) = \mathbf{x} \tag{24}$$

for which we are able to vary the location $p \leq q$ of an invariant initial condition $\mathbf{y}(p) = \mathbf{x}$. Suppose the sampled data is only available at the fixed output $t = q$ of the process, meaning that the training data is given by a set of the form

$$\{y_i := y(q; p_i, \mathbf{x}) | p_i \leq q\}_i. \tag{25}$$

In this way, an InImNet learns the dynamical system itself, as the initial conditions vary, from the outputs alone. Apply this to the running loss in Theorem 2 by, for a differentiable cost function $C \colon \mathbb{R}^N \times \mathbb{R}^N \to \mathbb{R}$, defining

$$\mathcal{R}(t, z, \boldsymbol{\theta}) = \mathcal{R}(t, z) := \sum_i C(z(q; p, \mathbf{z}(t; p, \mathbf{x})), \mathbf{y}_i) \delta_{t, p_i}. \tag{26}$$

Then the $p$-th invariant imbedding coefficient is

$$[\nabla_{\mathbf{z}} \mathcal{R}](p, \mathbf{x}) = \sum_i \delta_{p, p_i} \cdot [\nabla C_{\mathbf{z}}](z(q; p, \mathbf{x}), \mathbf{y}_i) \cdot \nabla_{\mathbf{x}} \mathbf{z}(q; p, \mathbf{x}). \tag{27}$$

The term $\delta_{p, p_i}$ features intrinsically in Neural ODE architectures as only a single depth $p$ is considered in a given system. We implement (26) recursively (cf. Chen et al., 2018, Figure 2) so that the discrete architecture for the adjoint reads

$$\mathbf{\Lambda}(p_i, \mathbf{x}) = \mathbf{\Lambda}(p_{i+1}, \mathbf{x}) + [\nabla_{\mathbf{z}} \mathcal{R}](p_i, \mathbf{x})$$
$$- (p_i - p_{i+1})[\nabla_{\mathbf{x}} \mathbf{\Lambda}(p_i, \mathbf{x}) \cdot \mathbf{\Phi}(p_i, \mathbf{x})) + \nabla_{\mathbf{x}} \mathbf{f}(p_i, \mathbf{x}, \boldsymbol{\theta}(p_i))^{\mathrm{T}} \cdot \mathbf{\Lambda}(p_{i+1}, \mathbf{x})]. \tag{28}$$

The forward pass is executed exactly as in §3.1.

## D    THE ADJOINT METHOD IN APPLICATION

### D.1    BACKPROPAGATION AND THE ADJOINT STATE

As touched on in the introduction, the adjoint method, in optimal control, is a natural consequence of the Euler–Lagrange equations (which we explicitly derive in §E.3). To outline the

method, one postulates a multiplier function $\boldsymbol{\lambda}(t; p, \mathbf{x})$, the 'Lagrange multiplier' or 'adjoint', as in §E.2. One derives the initial value of the adjoint, at the endpoint $t = q$, to be the loss gradient $\boldsymbol{\lambda}(q; p, \mathbf{x}) = [\nabla_{\mathbf{z}}\mathcal{T}](\mathbf{z}(q; p, \mathbf{x}))$. On the other hand, the $t$-varying dynamics is seen to follow the dynamical system given by (47). Integrating (47) backward from $t = q$ to $t = p$ may be interpreted as the backpropagation of losses throughout the system. Indeed, using routines such as those describe below in §D.2 the loss gradients with respect to parameters may be explicitly derived.

The interpretation of the adjoint as backpropagated losses was illuminated, in the context of deep neural networks by Chen et al. (2018). They derive the backward dynamical system (47) for $\boldsymbol{\lambda}$ using the chain rule over infinitesimal steps. This illumination makes clear that the adjoint method is the direct generalisation of the traditional backpropagation method.

In another twist in the tale, the derivation of the adjoint method also leads to a natural first-order optimality condition; we deduce this in (43). This theoretical tool gives an option for single-sweep optimisation by explicitly solving (43) for the parameters via the initial value problem (47). Moreover, this equation permits one to determine continuously varying parameters $\boldsymbol{\theta}(t)$. Such an investigation is conducted by Vialard et al. (2020).

In the next section we describe the formulas required to determine loss gradients with respect to parameters in the case when the parameter vector $\boldsymbol{\theta} = \boldsymbol{\theta}(t)$ is not dependent on $t$.

## D.2  AUGMENTED STATE VARIABLES FOR CONSTANT PARAMETERS

Updates to $t$-varying parameter controls $\boldsymbol{\theta}(t; p, \mathbf{x})$ are a hot subject of research (see Vialard et al., 2020; Massaroli et al., 2020, for example). The original Neural ODEs (Chen et al., 2018) operated by assuming that $\boldsymbol{\theta} = \boldsymbol{\theta}(t; p, \mathbf{x}) \in \mathbb{R}^M$ was a constant. It was then possible to recover loss gradients for the components of $\boldsymbol{\theta}$ by augmenting the system and running the adjoint method (Chen et al., 2018, §B.2). A similar trick works in our context, with some quite remarkable consequences.

Consider an augmented state vector composed of $\mathbf{z}$, $t$ and $\theta$, whereby the dynamics in (1) becomes

$$\frac{d}{dt} \begin{bmatrix} \mathbf{z} \\ t \\ \boldsymbol{\theta} \end{bmatrix} = \begin{bmatrix} \mathbf{f}(t, \mathbf{z}, \boldsymbol{\theta}) \\ 1 \\ \mathbf{0} \end{bmatrix} \tag{29}$$

with $\mathbf{z}(p) = \mathbf{x}$, $t = p$ and $\boldsymbol{\theta} = \boldsymbol{\theta}$ as the initial condition. Working through the invariant imbedding procedure, the forward process (Theorem 1) for $\mathbf{z}$ is preserved. But the backward process (Theorem 2) is revised by replacing $\boldsymbol{\Lambda}$ with the (new) components $\boldsymbol{\Lambda}^{(\mathbf{z})}$, $\boldsymbol{\Lambda}^{(t)}$ and $\boldsymbol{\Lambda}^{(\boldsymbol{\theta})}$. The first, $\boldsymbol{\Lambda}^{(\mathbf{z})}$, satisfies Theorem 2 independently of $\boldsymbol{\Lambda}^{(t)}$ and $\boldsymbol{\Lambda}^{(\boldsymbol{\theta})}$. Moreover, the assumption that $\boldsymbol{\theta} = \boldsymbol{\theta}(t)$ is constant in $t$ implies that $[\nabla_{\mathbf{z}}\mathbf{f}](p, \mathbf{x}, \boldsymbol{\Psi}(p, \mathbf{x})) = \nabla_{\mathbf{x}}\boldsymbol{\Phi}(p, \mathbf{x})$ so we deduce the new backward equation

$$\nabla_p\boldsymbol{\Lambda}^{(\mathbf{z})}(p, \mathbf{x}) = -(\nabla_{\mathbf{x}}\boldsymbol{\Lambda}^{(\mathbf{z})}(p, \mathbf{x}) \cdot \boldsymbol{\Phi}(p, \mathbf{x}) + \nabla_{\mathbf{x}}\boldsymbol{\Phi}(p, \mathbf{x})^{\mathrm{T}} \cdot \boldsymbol{\Lambda}^{(\mathbf{z})}(p, \mathbf{x}) + [\nabla_{\mathbf{z}}\mathcal{R}](p, \mathbf{x}, \boldsymbol{\Psi}(p, \mathbf{x}))). \tag{30}$$

The term $\boldsymbol{\Lambda}^{(\boldsymbol{\theta})}(p, \mathbf{x})$, corresponding to the gradient of the terminal loss with respect to the constant parameters $\boldsymbol{\theta}$, assumes the initial value

$$\boldsymbol{\Lambda}^{(\boldsymbol{\theta})}(q, \mathbf{x}) = \mathbf{0} \tag{31}$$

for whom the $p$-evolution (10) becomes

$$\nabla_p\boldsymbol{\Lambda}^{(\boldsymbol{\theta})}(p, \mathbf{x}) = -(\nabla_{\mathbf{x}}\boldsymbol{\Lambda}^{(\boldsymbol{\theta})}(p, \mathbf{x}) \cdot \boldsymbol{\Phi}(p, \mathbf{x}) + \boldsymbol{\Phi}_{\boldsymbol{\theta}}(p, \mathbf{x})^{\mathrm{T}} \cdot \boldsymbol{\Lambda}^{(\mathbf{z})}(p, \mathbf{x}) + [\nabla_{\boldsymbol{\theta}}\mathcal{R}](p, \mathbf{x}, \boldsymbol{\Psi}(p, \mathbf{x}))), \tag{32}$$

dependant on the simultaneous solution of $\boldsymbol{\Lambda}^{(\mathbf{z})}(p, \mathbf{x})$.

For the $t$-term let us additionally assume that $\mathbf{f}(t, \mathbf{z}, \boldsymbol{\theta}) = \mathbf{f}(\mathbf{z}, \boldsymbol{\theta})$ and $\mathcal{R}(t, \mathbf{z}, \boldsymbol{\theta}) = \mathcal{R}(\mathbf{z}, \boldsymbol{\theta})$. This implies that $\nabla_t\mathbf{f}(p, \mathbf{x}, \boldsymbol{\Psi}(p, \mathbf{x})) = \nabla_{\mathbf{x}}\boldsymbol{\Phi}(p, \mathbf{x}) \cdot \boldsymbol{\Phi}(p, \mathbf{x})$. Then $\boldsymbol{\Lambda}^{(t)}(p, \mathbf{x})$, the gradient of the terminal loss with respect to $t = p$, assumes the initial value

$$\boldsymbol{\Lambda}^{(t)}(q, \mathbf{x}) = \langle \boldsymbol{\Lambda}^{(\mathbf{z})}(q, \mathbf{x}), \boldsymbol{\Phi}(q, \mathbf{x}) \rangle \tag{33}$$

and the $p$-evolution follows

$$\begin{aligned} \nabla_p\boldsymbol{\Lambda}^{(t)}(p, \mathbf{x}) = -[\nabla_{\mathbf{x}}\boldsymbol{\Lambda}^{(t)}(p, \mathbf{x}) \cdot \boldsymbol{\Phi}(p, \mathbf{x}) + (\nabla_{\mathbf{x}}\boldsymbol{\Phi}(p, \mathbf{x}) \cdot \boldsymbol{\Phi}(p, \mathbf{x}))^{\mathrm{T}} \cdot \boldsymbol{\Lambda}^{(\mathbf{z})}(p, \mathbf{x}) \\ + \nabla_{\mathbf{x}}\mathcal{R}(p, \mathbf{x}, \boldsymbol{\Psi}(p, \mathbf{x})) \cdot \boldsymbol{\Phi}(p, \mathbf{x})], \quad \end{aligned} \tag{34}$$

dependant on the simultaneous solution of $\mathbf{\Lambda}^{(\mathbf{z})}(p, \mathbf{x})$. This last identity is particularly exciting. It enables one to compute the loss with respect to $p$, the depth, itself. Given the explication of this $p$-variable in our Invariant Imbedding Networks, one may thus plot this loss as a function of $p$ and isolate the minima for any given training run. Observing the flow of such minima provides great insight to the optimal depth that a network should take.

# E    DERIVING THE INVARIANT IMBEDDING SOLUTION

Here we give proofs of Theorems 1, 2 and 3 which constitute an invariant imbedding solution for a non-linear, vector-valued Bolza problem. Consider the minimisation problem described in §2.1 for the ODE system (1). We maintain the hypotheses stated in Theorem 1 throughout.

## E.1    VARIATIONS IN THE CONTROL

We explore the consequences of the calculus of variations (Liberzon, 2012, Ch. 2). This provides a mechanism to find an *optimal control* $\boldsymbol{\theta}$; that is, $\boldsymbol{\theta}$ provides a global minimum for $\mathcal{J}(\boldsymbol{\theta}) \leq \mathcal{J}(\tilde{\boldsymbol{\theta}})$ over all piecewise continuous controls $\tilde{\boldsymbol{\theta}}$. For $\varepsilon \in \mathbb{R}$ considered close to 0, we consider controls perturbed from the optimal control $\boldsymbol{\theta}$ by

$$\boldsymbol{\theta}(t; p, \mathbf{x}) + \varepsilon \boldsymbol{\nu}(t; p, \mathbf{x}) \tag{35}$$

for an admissible perturbation $\boldsymbol{\nu}$. Following the system (1), the control in (35) determines the state (or trajectory)

$$\mathbf{z}(t; p, \mathbf{x}) + \varepsilon \boldsymbol{\eta}(t; p, \mathbf{x}) + o(\varepsilon) \tag{36}$$

where $\boldsymbol{\eta}$ is a corresponding perturbation subject to

$$\boldsymbol{\eta}(p; p, \mathbf{x}) = 0 \tag{37}$$

so that (36) agrees with $\mathbf{z}$ at $t = p$. To minimise the cost, we consider the Taylor expansion of $\mathcal{J}(\boldsymbol{\theta} + \varepsilon \boldsymbol{\nu})$ with respect to $\varepsilon$. We call the linear term *the first variation* of $\mathcal{J}$ at $\boldsymbol{\theta}$ which may be defined by the Gateaux derivative:

$$\delta \mathcal{J}|_{\boldsymbol{\theta}}(\boldsymbol{\nu}) := \lim_{\varepsilon \to 0} \frac{\mathcal{J}(\boldsymbol{\theta} + \varepsilon \boldsymbol{\nu}) - \mathcal{J}(\boldsymbol{\theta})}{\varepsilon} \tag{38}$$

for admissible perturbations $\boldsymbol{\nu}$. For the cost function given in (4), the first variation is equal to

$$\delta \mathcal{J}|_{\boldsymbol{\theta}}(\boldsymbol{\nu}) = \int_p^q \langle [\nabla_{\boldsymbol{\theta}} \mathcal{R}](t, \mathbf{z}, \boldsymbol{\theta}), \boldsymbol{\nu}(t; p, \mathbf{x}) \rangle dt + \int_p^q \langle [\nabla_{\mathbf{z}} \mathcal{R}](t, \mathbf{z}, \boldsymbol{\theta}), \boldsymbol{\eta}(t; p, \mathbf{x}) \rangle dt$$
$$+ \langle [\nabla_{\mathbf{z}} \mathcal{T}](\mathbf{z}(q; p, \mathbf{x})), \boldsymbol{\eta}(q; p, \mathbf{x}) \rangle. \tag{39}$$

The first-order, necessary condition for optimality (Liberzon, 2012, §1.3.2) is satisfied whenever

$$\delta \mathcal{J}|_{\boldsymbol{\theta}}(\boldsymbol{\nu}) = 0. \tag{40}$$

An auxiliary identity for the manipulation of (39) is obtained by evaluating $\dot{\boldsymbol{\eta}}$. Differentiating (36) with respect to $\varepsilon$ about $\varepsilon = 0$, both directly and via (1), implies that

$$\dot{\boldsymbol{\eta}}(t; p, \mathbf{x}) = [\nabla_{\mathbf{z}} \mathbf{f}](t, \mathbf{z}, \boldsymbol{\theta}) \cdot \boldsymbol{\eta}(t; p, \mathbf{x}) + [\nabla_{\boldsymbol{\theta}} \mathbf{f}](t, \mathbf{z}, \boldsymbol{\theta}) \cdot \boldsymbol{\nu}(t; p, \mathbf{x}). \tag{41}$$

## E.2    LAGRANGE MULTIPLIERS

Introduce $\boldsymbol{\lambda}(t) \in \mathbb{R}^N$, an arbitrary vector-valued function for $t \in [p, q]$ to be specified forthwith. This is the *Lagrange multiplier*, called so as its namesake used it to multiply Equation (41) and insert it into the first variation (39). We subsequently obtain

$$\delta \mathcal{J}|_{\boldsymbol{\theta}}(\boldsymbol{\nu}) = \int_p^q \langle [\nabla_{\boldsymbol{\theta}} \mathcal{R}](t, \mathbf{z}, \boldsymbol{\theta}) + [\nabla_{\boldsymbol{\theta}} \mathbf{f}](t, \mathbf{z}, \boldsymbol{\theta})^{\mathrm{T}} \cdot \boldsymbol{\lambda}(t), \boldsymbol{\nu}(t; p, \mathbf{x}) \rangle dt$$

$$+ \int_p^q \langle \boldsymbol{\lambda}(t), [\nabla_{\mathbf{z}} \mathbf{f}](t, \mathbf{z}, \boldsymbol{\theta}) \cdot \boldsymbol{\eta}(t; p, \mathbf{x}) - \dot{\boldsymbol{\eta}}(t; p, \mathbf{x}) \rangle dt$$

$$+ \int_p^q \langle [\nabla_{\mathbf{z}} \mathcal{R}](t, \mathbf{z}, \boldsymbol{\theta}), \boldsymbol{\eta}(t; p, \mathbf{x}) \rangle dt + \langle [\nabla_{\mathbf{z}} \mathcal{T}](\mathbf{z}(q; p, \mathbf{x})), \boldsymbol{\eta}(q; p, \mathbf{x}) \rangle. \tag{42}$$

We obtain an optimality condition by making a choice of $\boldsymbol{\lambda}(t) = \boldsymbol{\lambda}(t; p, \mathbf{x})$ such that

$$[\nabla_{\boldsymbol{\theta}}\mathcal{R}](t, \mathbf{z}, \boldsymbol{\theta}) + [\nabla_{\boldsymbol{\theta}}\mathbf{f}](t, \mathbf{z}, \boldsymbol{\theta})^{\mathrm{T}} \cdot \boldsymbol{\lambda}(t; p, \mathbf{x}) = 0 \tag{43}$$

for all $t \in [q, p]$; cf. (Vialard et al., 2020, (3.3)) where their "$\mathbf{p}_i$" is equal to our $-\boldsymbol{\lambda}$. The first variation (39) simplifies accordingly:

$$\delta\mathcal{J}|_{\boldsymbol{\theta}}(\boldsymbol{\nu}) = \int_p^q \langle \boldsymbol{\lambda}(t; p, \mathbf{x}), [\nabla_{\mathbf{z}}\mathbf{f}](t, \mathbf{z}, \boldsymbol{\theta}) \cdot \boldsymbol{\eta}(t; p, \mathbf{x}) - \dot{\boldsymbol{\eta}}(t; p, \mathbf{x}) \rangle dt$$

$$+ \int_p^q \langle [\nabla_{\mathbf{z}}\mathcal{R}](t, \mathbf{z}, \boldsymbol{\theta}), \boldsymbol{\eta}(t; p, \mathbf{x}) \rangle dt + \langle [\nabla_{\mathbf{z}}\mathcal{T}](\mathbf{z}(q; p, \mathbf{x})), \boldsymbol{\eta}(q; p, \mathbf{x}) \rangle. \tag{44}$$

At this point our method of invariant imbedding diverges from the classical method of Euler and Lagrange. We give a brief aside here to review the Euler–Lagrange equations as the comparison is illuminating; see §E.3. We also direct the reader to (Liberzon, 2012, §3.3-3.4) in which the classical route is explored.

### E.3 THE EULER–LAGRANGE EQUATIONS

The Euler–Lagrange equations reveal how the adjoint method of Chen et al. (2018), see also (Vialard et al., 2020, §3.3), will correspond to our invariant imbedding formulation. The derivation follows from integrating (44) by parts with respect to $\dot{\boldsymbol{\eta}}$. One derives

$$\delta\mathcal{J}|_{\boldsymbol{\theta}}(\boldsymbol{\nu}) = \int_p^q \langle [\nabla_{\mathbf{z}}\mathbf{f}](t, \mathbf{z}, \boldsymbol{\theta})^{\mathrm{T}} \cdot \boldsymbol{\lambda}(t; p, \mathbf{x}) + [\nabla_{\mathbf{z}}\mathcal{R}](t, \mathbf{z}, \boldsymbol{\theta}) + \dot{\boldsymbol{\lambda}}(t; p, \mathbf{x}), \boldsymbol{\eta}(t; p, \mathbf{x}) \rangle dt$$

$$+ \langle [\nabla_{\mathbf{z}}\mathcal{T}](\mathbf{z}(q; p, \mathbf{x})) - \boldsymbol{\lambda}(q; p, \mathbf{x}), \boldsymbol{\eta}(q; p, \mathbf{x}) \rangle. \tag{45}$$

noting that the initial value of the perturbation is $\boldsymbol{\eta}(p; p, \mathbf{x}) = 0$ by assumption. We simplify the constant term by imposing the initial condition on $\boldsymbol{\lambda}$:

$$\boldsymbol{\lambda}(q; p, \mathbf{x}) = [\nabla_{\mathbf{z}}\mathcal{T}](\mathbf{z}(q; p, \mathbf{x})). \tag{46}$$

Since (45) equates to 0 by the first order optimality condition (40) for all admissible perturbations $\boldsymbol{\eta}$, the fundamental lemma of the calculus of variations implies

$$\dot{\boldsymbol{\lambda}}(t; p, \mathbf{x}) = -([\nabla_{\mathbf{z}}\mathbf{f}](t, \mathbf{z}, \boldsymbol{\theta})^{\mathrm{T}} \cdot \boldsymbol{\lambda}(t; p, \mathbf{x}) + [\nabla_{\mathbf{z}}\mathcal{R}](t, \mathbf{z}, \boldsymbol{\theta})). \tag{47}$$

Equations (46) and (47) constitute the continuous analogy to backpropagation known as the "adjoint method" by Chen et al. (2018). Its formulation is thus encoded in (44) and so the forthcoming method of invariant imbedding provides a direct alternative to this backpropagation approach.

### E.4 THE INVARIANT IMBEDDING METHOD

For each $0 \leq \Delta \leq q - p$ consider the family of subintervals $[p + \Delta, q]$ of $[p, q]$. Over these intervals, we imbed the systems $\mathbf{z}(t; p + \Delta, \mathbf{x})$ into $\mathbf{z}(t; p, \mathbf{x})$ whilst keeping $\mathbf{x}$, the input, invariant. This results in the partial differential equations (51) and (52) in which changes in $p$ are equated to changes with respect to $\mathbf{x}$.

To derive these equations, we first note that over each interval the systems follow (1), which may be expressed as a finite difference equation:

$$\mathbf{z}(t; p, \mathbf{x}) = \mathbf{z}(t + \Delta; p, \mathbf{x}) - \mathbf{f}(t, \mathbf{z}(t; p, \mathbf{x}), \boldsymbol{\theta}(t; p, \mathbf{x}))\Delta + o(\Delta) \tag{48}$$

for $p \leq t \leq q - \Delta$. We then formally extend $\mathbf{z}(t; p + \Delta, \mathbf{x})$ to $t = p$ via (48) so that

$$\mathbf{z}(p; p + \Delta, \mathbf{x}) = \mathbf{x} - \mathbf{f}(p, \mathbf{z}(p; p + \Delta, \mathbf{x}), \boldsymbol{\theta}(p; p + \Delta, \mathbf{x}))\Delta + o(\Delta). \tag{49}$$

As such, the imbedding of systems over $[p + \Delta, q] \subset [p, q]$ is explicitly defined by

$$\mathbf{z}(t; p + \Delta, \mathbf{x}) = \mathbf{z}(t; p, \mathbf{x} - \mathbf{f}(p, \mathbf{z}(p; p + \Delta, \mathbf{x}), \boldsymbol{\theta}(p; p + \Delta, \mathbf{x}))\Delta + o(\Delta)) \tag{50}$$

for all $p \leq t \leq q$. To compute the partial derivative $\nabla_p \mathbf{z} = \partial \mathbf{z}/\partial p$, consider the difference $\mathbf{z}(t; p + \Delta, \mathbf{x}) - \mathbf{z}(t; p, \mathbf{x})$ in which $\mathbf{x}$ remains invariant. Substituting (50), dividing by $\Delta$ and taking the limit $\Delta \to 0^+$ one obtains *the invariant imbedding identity for the trajectory*:

$$\nabla_p \mathbf{z}(t; p, \mathbf{x}) = -\nabla_{\mathbf{x}}\mathbf{z}(t; p, \mathbf{x}) \cdot \mathbf{f}(p, \mathbf{x}, \boldsymbol{\theta}(p; p, \mathbf{x})). \tag{51}$$

Mutatis mutandis, one derives *the invariant imbedding identity for the control*:

$$\nabla_p \boldsymbol{\theta}(t; p, \mathbf{x}) = -\nabla_{\mathbf{x}}\boldsymbol{\theta}(t; p, \mathbf{x}) \cdot \mathbf{f}(p, \mathbf{x}, \boldsymbol{\theta}(p; p, \mathbf{x})). \tag{52}$$

These equations express a fundamental translational invariance property of both the optimal control and the corresponding trajectory. This proves the basic identity in Theorem 1.

## E.5 INVARIANT IMBEDDING OF THE LAGRANGE MULTIPLIER

We extend the invariant imbedding relations (51) and (52) to the Lagrange multiplier $\boldsymbol{\lambda}(t; p, \mathbf{x})$. The extension is obtained by taking the gradient of (43) with respect to $\nabla_p = \partial/\partial p$ and $\nabla_{\mathbf{x}}$. Adding the resulting expressions together and using (51) and (52) to replace the terms $\nabla_p \mathbf{z}(t; p, \mathbf{x})$ and $\nabla_p \boldsymbol{\theta}(t; p, \mathbf{x})$ we derive the identity

$$[\nabla_{\boldsymbol{\theta}} \mathbf{f}](t, \mathbf{z}, \boldsymbol{\theta})^{\mathrm{T}} \cdot (\nabla_p \boldsymbol{\lambda}(t; p, \mathbf{x}) + \nabla_{\mathbf{x}} \boldsymbol{\lambda}(t; p, \mathbf{x}) \cdot \mathbf{f}(p, \mathbf{x}, \boldsymbol{\theta}(p; p, \mathbf{x}))). \tag{53}$$

For non-trivial dependence of $\mathbf{f}$ on the control $\boldsymbol{\theta}$ we assume that $[\nabla_{\boldsymbol{\theta}}]\mathbf{f}(t, \mathbf{z}, \boldsymbol{\theta})$ is not identically 0 implying *the invariant imbedding identity for the Lagrange multiplier,* $\boldsymbol{\lambda}$:

$$\nabla_p \boldsymbol{\lambda}(t; p, \mathbf{x}) = -\nabla_{\mathbf{x}} \boldsymbol{\lambda}(t; p, \mathbf{x}) \cdot \mathbf{f}(p, \mathbf{x}, \boldsymbol{\theta}(p; p, \mathbf{x})). \tag{54}$$

These equations express a fundamental translational invariance property of both the optimal control and the corresponding trajectory.

The coefficients $\mathbf{f}(p, \mathbf{x}, \boldsymbol{\theta}(p; p, \mathbf{x}))$ and $\boldsymbol{\theta}(p; p, \mathbf{x})$ in (51) and (52) depend on the variables $(p, \mathbf{x})$ alone, independently of $t$. It is thus sensible to introduce the terminology

$$\boldsymbol{\Phi}(p, \mathbf{x}) := \mathbf{f}(p, \mathbf{x}, \boldsymbol{\theta}(p; p, \mathbf{x})) \in \mathbb{R}^N; \tag{55}$$

$$\boldsymbol{\Psi}(p, \mathbf{x}) := \boldsymbol{\theta}(p; p, \mathbf{x}) \in \mathbb{R}^M. \tag{56}$$

## E.6 FIRST ORDER OPTIMALITY WITH INVARIANT IMBEDDING

Here we derive an auxiliary optimality condition from the first variation (44) subject to the invariant imbedding relations (51), (52) and (54). Recall that (44) is equivalent to the adjoint-state equation of Chen et al. (2018), seen by application of the Euler–Lagrange equations; §E.3. The theory we develop here should thus also be seen as analogue to the standard backpropagation method.

The first variation (44) is equal to 0 by (40). One may take its gradient with respect to $\nabla_p = \partial/\partial p$ using Leibniz' rule and apply $\boldsymbol{\eta}(p; p, \mathbf{x}) = 0$; moreover, each occurrence of $\nabla_p \mathbf{z}$, $\nabla_p \boldsymbol{\theta}$ and $\nabla_p \boldsymbol{\lambda}$ may be substituted with their respective invariant imbedding relation. Then take the sum of the resulting equation with the gradient of (44) with respect to $\nabla_{\mathbf{x}}$ scaled by the term $\boldsymbol{\Phi}(p, \mathbf{x})$. One thus obtains the general form of the first-variation auxiliary equation:

$$\langle [\nabla_{\mathbf{z}} \mathcal{T}](\mathbf{z}(q; p, \mathbf{x})), \nabla_p \boldsymbol{\eta}(q; p, \mathbf{x}) + \nabla_{\mathbf{x}} \boldsymbol{\eta}(q; p, \mathbf{x}) \cdot \boldsymbol{\Phi}(p, \mathbf{x}) \rangle + \langle \boldsymbol{\lambda}(p; p, \mathbf{x}), \dot{\boldsymbol{\eta}}(p; p, \mathbf{x}) \rangle$$

$$\int_p^q \langle [\nabla_{\mathbf{z}} \mathbf{f}](t, \mathbf{z}, \boldsymbol{\theta})^{\mathrm{T}} \cdot \boldsymbol{\lambda}(t; p, \mathbf{x}) + [\nabla_{\mathbf{z}} \mathcal{R}](t, \mathbf{z}, \boldsymbol{\theta}), \nabla_p \boldsymbol{\eta}(t; p, \mathbf{x}) + \nabla_{\mathbf{x}} \boldsymbol{\eta}(t; p, \mathbf{x}) \cdot \boldsymbol{\Phi}(p, \mathbf{x}) \rangle dt$$

$$+ \int_p^q \langle \boldsymbol{\lambda}(t; p, \mathbf{x}), \nabla_p \dot{\boldsymbol{\eta}}(t; p, \mathbf{x}) + \nabla_{\mathbf{x}} \dot{\boldsymbol{\eta}}(t; p, \mathbf{x}) \cdot \boldsymbol{\Phi}(p, \mathbf{x}) \rangle dt = 0. \tag{57}$$

The derivation of (57) requires the multiplication of higher order tensors. This is readily completed using Einstein's summation convention. Note that before specifying our choice of $\boldsymbol{\eta}$ there is no stipulation that it should adhere to an invariant imbedding rule of the form $\nabla_p \boldsymbol{\eta} = -\nabla_{\mathbf{x}} \boldsymbol{\eta} \cdot \boldsymbol{\Phi}$.

## E.7 SPECIFYING THE VARIATIONS TO SOLVE FOR THE LAGRANGE MULTIPLIER

The auxiliary optimality condition (57) holds for all admissible perturbations $(\boldsymbol{\nu}, \boldsymbol{\eta})$. We now make specific choices to derive an initial condition for $\boldsymbol{\lambda}(t; p, \mathbf{x})$ at $t = p$. For $1 \le j \le N$ define the column vector

$$\boldsymbol{\eta}_j(t; p, \mathbf{x}) := ((t - p)\delta_{ij})_i. \tag{58}$$

This perturbation satisfies $\boldsymbol{\eta}_j(p; p, \mathbf{x}) = \mathbf{0}$, as required, and has derivatives given by $\dot{\boldsymbol{\eta}}_j = -\nabla_p \boldsymbol{\eta}_j = (\delta_{ij})_i$ and $\nabla_p \dot{\boldsymbol{\eta}}_j = \mathbf{0}$; whilst all gradients with respect to $\nabla_{\mathbf{x}}$ satisfy $\nabla_{\mathbf{x}} \boldsymbol{\eta}_j = \nabla_{\mathbf{x}} \dot{\boldsymbol{\eta}}_j = \mathbf{0}$. By substitution into (57) we show that

$$\boldsymbol{\lambda}(p; p, \mathbf{x}) = [\nabla_{\mathbf{z}} \mathcal{T}](\mathbf{z}(q; p, \mathbf{x})) - \int_q^p ([\nabla_{\mathbf{z}} \mathbf{f}](t, \mathbf{z}, \boldsymbol{\theta})^{\mathrm{T}} \cdot \boldsymbol{\lambda}(t; p, \mathbf{x}) + [\nabla_{\mathbf{z}} \mathcal{R}](t, \mathbf{z}, \boldsymbol{\theta})) dt. \tag{59}$$

Note that by the fundamental theorem of calculus, the initial condition in (59) is equivalent to the adjoint equation of Chen et al. (2018).

The remaining results are then derived as follows. Theorem 2 follows directly from (59). Take the derivatives $\nabla_p \mathbf{\Lambda}(p, \mathbf{x})$ and $\nabla_{\mathbf{x}} \mathbf{\Lambda}(p, \mathbf{x})$ from the explicit formula in (59) and substitute the $\nabla_p$ terms with the invariant imbedding relations (51), (52) and (54). Letting $\nabla_{\mathbf{x}} \mathbf{\Lambda}(p, \mathbf{x})$ operate on $\mathbf{\Phi}(p, \mathbf{x})$ Theorem 2 follows by summing the two equations. The initial condition at $p = q$ follows immediately from (59). Similarly, Theorem 3 follows by specifying (40) at the endpoint $t = q$.

## F    FURTHER EXPERIMENTATION

### F.1    INIMNETS AND LEARNING DYNAMICAL SYSTEMS

A significant speed bump in the training of InImNet architectures is the computation of nested Jacobians $\nabla_{\mathbf{x}}$. This problem increases exponentially with depth. We are able to demonstrate successful results with low depth implementations. We also provide elementary mechanisms to circumvent the issue; see §3.2.

Time series problems require vastly deep InImNets given the training data is spread along a series of times $t$ (or rather depths $p$). This is the case when applying InImNets to model ODE dynamics directly. But it is the immediate alternative–to replace the Jacobians with numerical gradients as in §3.2–that draws out a more interesting problem. The stability due to the initial divergence of the numerical gradients is related to 'Lyapunov exponents' (Lyapunov, 1992) that measure the growth rates of generic perturbations. The simulation and optimisation of ODEs by neural networks with perturbed initial conditions, in particular InImNets, is a deep one and the subject of immediate ongoing research.

We nevertheless are able to construct a rigorous architecture to handle time series data–see §C–and demonstrate its functionality on a sufficiently small experiment.

### F.2    A PROJECTILE MOTION REGRESSION PROBLEM

InImNets can be trained to model a unique brand of time series observations. We depict this with the example of projectile curves, see Figure 1. The data provided is not a sample along the $t$-varying shape of a given curve, but the single output of many curves initialised at different positions with the same velocity.

We simulate this via the discrete running cost training paradigm in §C. The system we consider is parameterised by a state variable $\mathbf{z} = [h, v]$, denoting vertical height and velocity, such that the $t$-axis is proportional to horizontal position. The ODE system dynamics is then of the form $\dot{\mathbf{z}}(t) = [v(t), -g]$ where $g$, gravity, is a positive scalar constant (which we take to equal 9.81). We choose to integrate $p$ (backward) over the interval $[p_{\min}, q] = [0, 1]$.

The architecture we use is the backward loss propagation described in Theorem 2 adapted to the training data by the time-series algorithm in §C. We implement a constant-in-$t$ training function $\mathbf{f}$ and further make use of the augmented adjoint routine described in §D.2 to update the parameters of $\mathbf{f}$. We run the experiment with a learning rate of 0.001 over 10 training epochs.

We operate the experiment at a resolution of points $p = 0, 0.25, 0.5, 0.75, 1$ with a sample size of just four. Using the automatic differentiation method for computation of Jacobians as per Equation (15), this is the maximum resolution that we are able to run without RAM availability being exhausted by growing Jacobian calculations–this is prototypical of the difficulties described in §F.1. As such, the InImNet performs poorly on this task in lieu of sufficient $p$-resolution, and hence training data. Nevertheless, this experiments suffices to prove the concept that InImNets may be used to solve a different genre of time-series problems, with further research demanded on the implementation of their nested derivatives.

We implement this experiment in a Google Colab notebook which is provided as a supplement to this article.

### F.3    EXTRAPOLATION OF OUTPUTS BEYOND OPTIMISED DEPTHS

To further demonstrate the ability of InImNets to produce meaningful representations at different layers, we have conducted experiments that extrapolate results for depths $p$ beyond $p_{\min}$. This

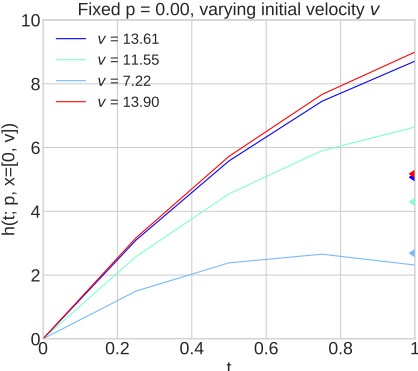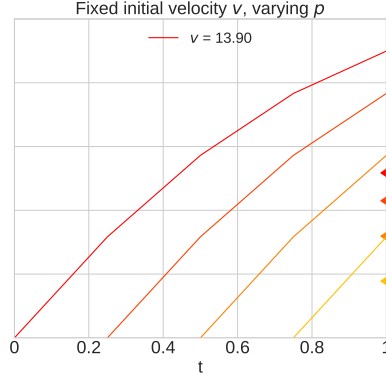

Figure 6: Plotted curves are given as training samples, where the InImNet's task, trained only on end points, is to identify the curves' height $h(t; p, \mathbf{x})$ at $t = q = 1$. The InImNet outputs are represented by correspondingly coloured triangular markers.

considers networks of greater depth, with training occurring at lower layers only. In our experimental architecture of §3.1 the weights are optimised independently: each $i$-th layer is parameterised by a distinct $\mathbf{\Phi}(p_i, \mathbf{x})$. Therefore we cannot perform extrapolation in such a setting. To compensate, we have introduced a variant InImNet architecture for both the rotating MNIST and the bouncing balls tasks where $\mathbf{\Phi}$ is identical for all layers; that is, for each $i$ we have $\mathbf{\Phi}(p_i, \mathbf{x}) = \mathbf{\Phi}(\mathbf{x})$. This is comparable to the $t$-invariant parameters used by Chen et al. (2018). In this architecture we take $p_{\min} = -3$ for bouncing balls and $p_{\min} = -4$ for the rotating MNIST. Otherwise, the rest of the parameters have not been changed from the original model hyperparameters (see §G).

With this architecture, we observe competitive performance for rotating MNIST–albeit worse than the original architecture varying $\mathbf{\Phi}$ over the layers in §4.1.1 and §G.1–as well as the impact of going beyond $p_{\min}$. Figure 5 shows the average performance depending on the frame sequence number. One can see that the accuracy decreases for higher layers. However, we can make quantitative observations about the rate of this loss change in different directions. For example, the loss is lost at a slower rate moving deeper into the network ($p = -5$) rather than shallower ($p = -3$).

The examples of extrapolated sequences from the testing set are given in Figure 7. It can be observed that the extrapolation in higher layers (such as -8) is accompanied by decrease in sample diversity across the sequence.

For the bouncing balls task, we see that coupling of the weights makes it more challenging to predict the balls' dynamics with the same hyperparameters as in §G.2, as expected. However, we demonstrate here that we can quantify and explain this discrepancy as the model gives us the insight into what it learnt. In Figure 7, while not successfully modelling the dynamics, the model converges to the 'best' strategy of erasing the sequence. Once we increase the layers, the same effect results in the reversal of the original sequence.

## F.4 EXPERIMENTS WITH CONVOLUTIONAL ARCHITECTURES

We demonstrate the ability of the InImNet architectures to be composed of larger convolutional layers, allowing for end-to-end training of an InImNet model. With the autoencoder architecture as in Vialard et al. (2020) to define an InImNet layer and repeat the experiments modelling bouncing balls as outlined in §4.1.2. For explicit details on the architecture of this model, see §G.3.

We report on performance results for $p_{\min} = \{-1, -2, -3\}$ in Figure 8 and give sample demonstrations for different layer numbers in Figure 9.

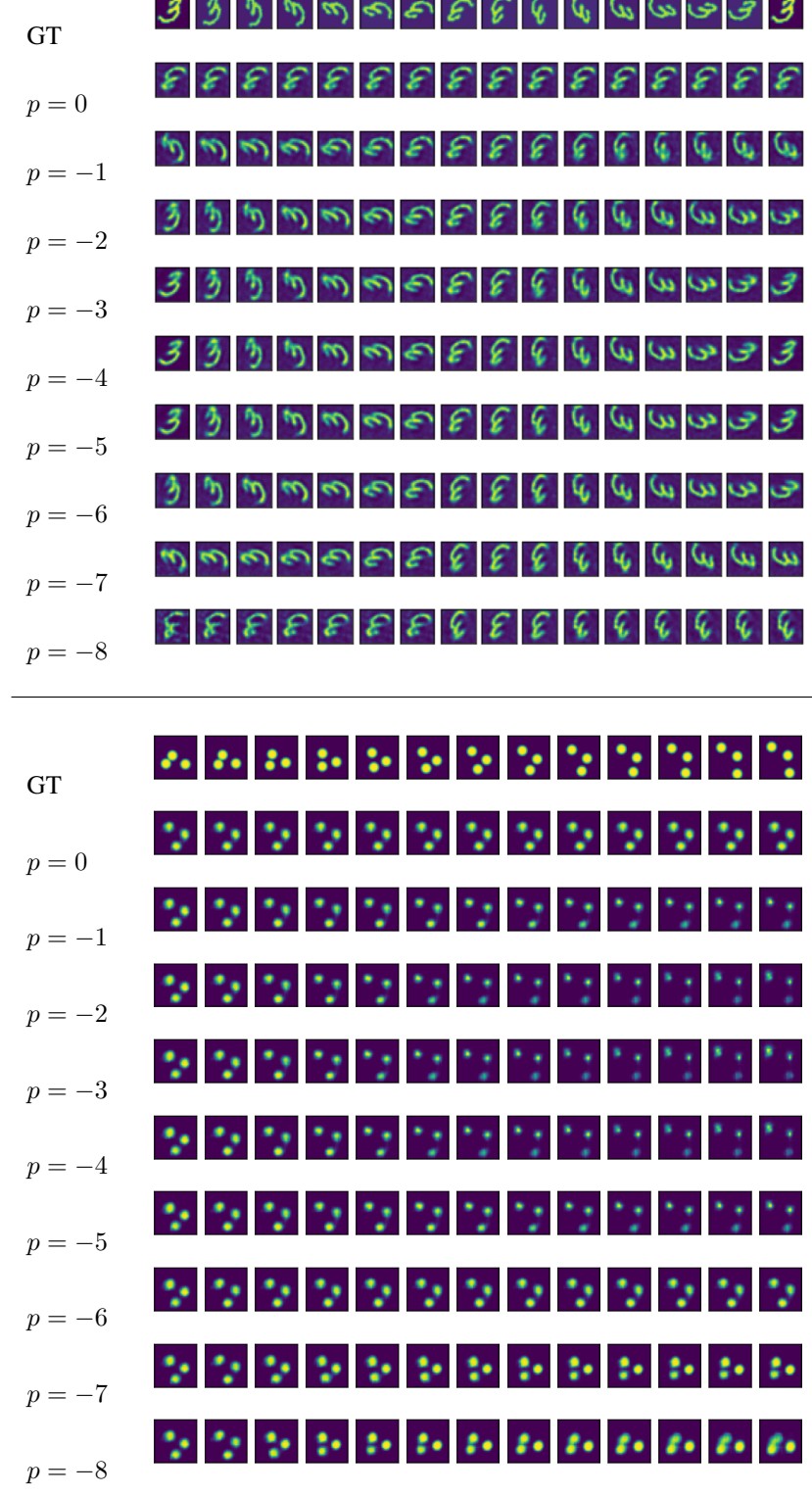

Figure 7: Extrapolated ouputs for the Rotating MNIST (above) and Bouncing Balls experiments. The models are optimised dependant on their output at level $p_{\min} = -4$ and $p_{\min} = -3$, respectively. One is able to assess performance at extrapolated depths $p \neq p_{\min}$ around these.

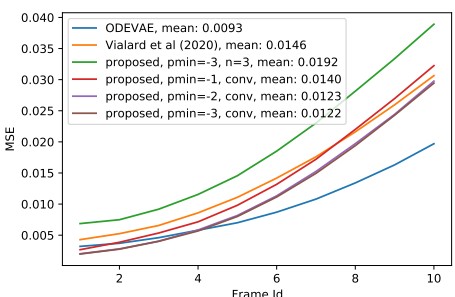

| $p_{\min}$ | MSE |
|---|---|
| InImNet, $p_{\min} = -1$ | 0.0140 |
| InImNet, $p_{\min} = -2$ | 0.0123 |
| InImNet, $p_{\min} = -3$ | 0.0122 |
| Vialard et al (2020) | 0.0146 |

Figure 8: Bouncing balls experiment. Left: reported MSE for the proposed InImNet and state-of-the-art methods, results from other methods are taken from Yıldız et al. (2019) and Vialard et al. (2020). Right: average time consumption per epoch.

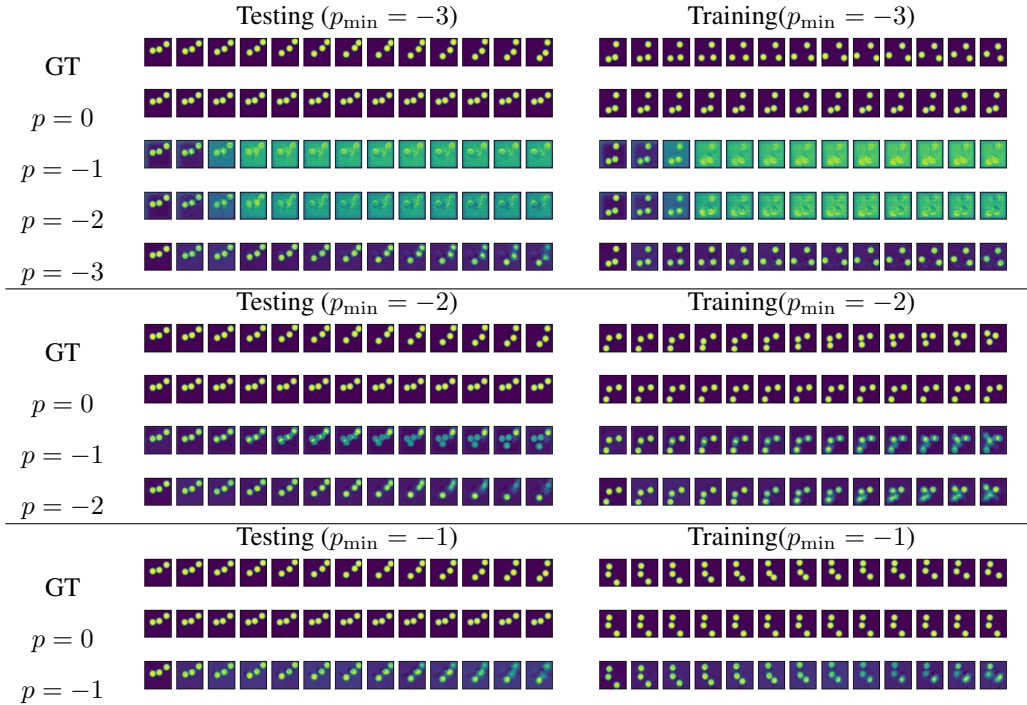

Figure 9: Bouncing balls experiment using a convolutional architecture; details of the architecture are given in Section G.3.

Equation (21) outlines the approximation we implement allowing us to avoid high memory costs during training whilst needing to evaluate a growing number of nested Jacobians. The aforementioned costs are associated with the recurrent computation of $\nabla_{\mathbf{x}} \mathbf{z}(t, p_i, \mathbf{x})$ which would require the computation of an $i$-th order gradient tree; this hold up is also discussed in §F.1. While it is still possible to use the exact solution–via automatic differentiation and gradient graphs–for some of the smaller problems with small $p_{\min}$, we use the approximation from Equation (21) in all discrete architecture experiments with high performance success and with acceptable memory costs.

## G    MODEL HYPERPARAMETERS

### G.1    ROTATING MNIST

We reproduce the experimental setting from Vialard et al. (2020); the experiments have been conducted in Tensorflow 2.6.0. The autoencoder's architecture, reimplemented in Tensorflow from the official pytorch code of Vialard et al. (2020), and parameters, including the batch size, replicate the same experimental set-up. We follow the discrete InImNet architecture described in Section 3.1, and we minimise the mean squared error loss between the ground truth and the outputs decoded at the layer $p_{\min}$ using backpropagation and gradient descent based optimisation. To enable the time series prediction, we append the time value to the latent representation of the autoencoder. The multilayer perceptron models use dropout regularisation for every layer except the last. The hyperparameters of the rotating MNIST model are listed in Table 2. All experiments have been run in a Google Colab GPU environment. Similar to Vialard et al. (2020), we denote as the inflation factor the size ratio between the intermediate and input MLP layers.

| Hyperparameter | Value |
|---|---|
| Optimiser | Adam |
| Batch size | 25 |
| Initial learning rate | 0.001 |
| Epochs | 500 |
| Dropout value | 0.3 |
| Inflation factor | 2 |
| Dimension of latent space | 20 |
| Learning rate schedule | 0.5 [decaying*] |

Table 2: Hyperparameters for the rotating MNIST experiment. (*) The learning rate decays exponentially at steps of 30 epochs.

### G.2    BOUNCING BALLS

The architecture of the autoencoder and experimental setting reimplements in Tensorflow 2.6.0 the one from the official code of Vialard et al. (2020), the rest of the considerations are identical to the ones given in G.1. As in the previous section, we concatenate the time parameter with the latent representation of the first three images in the autoencoder to obtain time series imbedding. The hyperparameters of the bouncing balls model are given in Table 3.

### G.3    CONVOLUTIONAL EXPERIMENTS

In our Bouncing-Balls-with-convolutional-architecture experiment, §F.4, every InImNet layer reimplements the fully-convolutional autoencoder from Vialard et al. (2020) in Tensorflow 2.6.0, with the exception of omitting the final layer's sigmoid (see the autoencoder description below). The dimensionality reduction is omitted as the input $\mathbf{x}$ and the outputs $\mathbf{z}(q; p, \mathbf{x})$ are $32 \times 32$ images. Following Vialard et al. (2020), we concatenate the first three images into the three channels of the autoencoder input, and add the fourth channel filled with the time value shaped $32 \times 32$. We obtain the prediction by taking the first channel of the autoencoder output. As in the previous experiments,

| Hyperparameter | Value |
| --- | --- |
| Optimiser | Adam |
| Batch size | 25 |
| Initial learning rate | 0.001 |
| Epochs | 100 |
| Dropout value | 0.3 |
| Inflation factor | 2 |
| Dimension of latent space | 50 |
| Learning rate schedule | 0.5 [decaying*] |

Table 3: Hyperparameters for the bouncing balls experiment. (*) The learning rate decays exponentially at steps of 30 epochs.

we optimise the mean squared error loss at the layer $p_{\min}$ and use backpropagation to optimise the parameters of all InImNet layers.

We calculate the Jacobians by flattening the values of $\mathbf{x}$ and $\mathbf{z}(q; p, \mathbf{x})$ into a vector of $32 \times 32 \times 4$ values. The overall structure of the autoencoder as implemented in Tensorflow Keras is as follows.

Encoder:

1. `Conv2D(16, (5, 5), strides=2, padding='same')`
2. `BatchNormalization()`
3. `ReLU()`
4. `Conv2D(32, (5, 5), strides=2, padding='same'))`
5. `BatchNormalization()`
6. `ReLU()`
7. `Conv2D(64, (5, 5), strides=2, padding='same')`
8. `ReLU()`

Decoder:

1. `Conv2DTranspose(128, (3, 3), strides=2, padding='same')`
2. `BatchNormalization()`
3. `ReLU()`
4. `Conv2DTranspose(64, (5, 5), strides=2, padding='same')`
5. `BatchNormalization()`
6. `ReLU()`
7. `Conv2DTranspose(32, (5, 5), strides=2, padding='same')`
8. `BatchNormalization()`
9. `ReLU()`
10. `Conv2DTranspose(4, (5, 5), padding='same'))`

The hyperparameters for the described convolutional model are listed in Table 4.

| Hyperparameter | Value |
| --- | --- |
| Optimiser | Adam |
| Batch size | 2 |
| Learning rate | 0.001 |
| Epochs | 20 |
| Learning rate schedule | Constant |

Table 4: Hyperparameters for the convolutional Bouncing balls experiment.

