# OpenReview forum: "Imbedding Deep Neural Networks"
_ICLR.cc/2022/Conference — ICLR 2022 Spotlight_

### Official Review · Reviewer_4Mn5 · 2021-10-29

**Correctness:** 4
**Technical Novelty And Significance:** 4
**Empirical Novelty And Significance:** Not applicable
**Recommendation:** 8
**Confidence:** 5

**Main Review:**

**Strong points:**
The idea is interesting and well motivated. The paper provides the derivations for both the forward pass and backward pass for the gradients w.r.t. a new parameter p. The methods section builds the model from the ground-up and it is easy to follow.

**Weak points:**

The results section is a bit underwhelming. The results demonstrate the bare minimum that a) the approach works, matches the performance of other models b) how the performance changes when we vary p_max.

In the introduction, the paper puts  a lot of emphasis on understanding and interpreting the network. With this motivation, I expected to see more analysis on the learned ODE and the relation between the learned trajectories z(…) and the trajectories in the data space (for example. similarly to figure 1); or the analysis what depth of the network is required to learn the bounce off the walls in the bouncing ball experiment. Perhaps instead network interpretation, the authors could highlight and experiment with other benefits of the model: 1) we can take the output of the network after every “depth” and decode it into a meaningful output 2) we can do smooth interpolation between the input and the output.

**Additional questions:**
What happens if we go beyond p_max in the experiments?

Do you actually see high memory cost to justify using the finite difference method that is as accurate as the chosen delta? All experiments use MLP with a few layers

Can we just vary the t_max rather than conditioning on p?

More details on the experiments need to be provided in the results section as well. Specifically, 1) How do you choose q? 2) In the results section, what does p_max represent? Why do you vary p_max, even though the model integrates on the interval [p, q]. Should it be p_min?


**Summary Of The Paper:**

The paper extends the Neural ODE model by introducing the dependency on the extra parameter p that can be interpreted as a control parameter or the “depth” of the network. The models allows to decode the network output by varying this parameter.


**Summary Of The Review:**

The paper has an interesting idea to condition the Neural ODE on the parameter p that can be interpreted as the “depth” of the network. The Results section does not explore the new architecture enough and provides only the basic results to demonstrate that the model works. I suggest the acceptance of the paper, but recommend the authors to expand the results section.

---

> ### Author Response · Authors · 2021-11-20
> **Response to referee (part 1)**
>
> We thank you for you suggestions. We have itemised specific the points below and answer them directly.
>
> 1. ''The results section is a bit underwhelming. The results demonstrate the bare minimum that a) the approach works, matches the performance of other models b) how the performance changes when we vary p_max.''
>
> Answer:
> We agree with this observation for the improvement of the paper. We added more experimental analysis (Appendix F and Section 4.2) on the following tasks: projectile motion regression problem with the continuous architecture (F.1); extrapolation beyond optimised depths (F.2 and 4.2); experiments on the bouncing balls task with fully convolutional architecture. Also, in the main paper (section 4.1) we added time figures for the proposed discrete model and state-of-the-art methods.
>
>
> 2. ''In the introduction, the paper puts a lot of emphasis on understanding and interpreting the network. With this motivation, I expected to see more analysis on the learned ODE and the relation between the learned trajectories z(...) and the trajectories in the data space (for example. similarly to figure 1)''
>
>
> Answer:
> Figure 1 is emphasised at the forefront to build an analogy to quickly explain the concept of an InImNet. We certainly agree with the referee that the modelling real dynamical ODEs with the proposed underlying ODE is of great interest, and one we have considered. In doing so we incurred some difficult stumbling blocks related to the memory consumption of InImNets. These lead to interesting mathematical paradigms which we hope to explore in further research, which we explain below (as well as in the manuscript).
>
> We have experimented with the time series set-up as described in figure 1. However, a fundamental difficulty is incurred: nested Jacobian calculation increases exponentially with depth. This is not such an issue for terminal loss optimisations as one can use shallower depths as prototypes. However, running cost problems necessarily propagate deep into the network to apply their loss values.
>
> We include an example experiment (Section F.2) which describes the maximum depth our RAM capacity can cope with. This is too limited to train successful networks.
>
> Nevertheless, we give an explicit architecture that describes such time-series problems; see Section D. And we expand upon the pitfalls in such investigations; see Section F.1. This includes the problem of numerical gradient solutions (which don't incur the RAM costs) becoming unstable.
>
> We provide a Google Colab notebook to exhibit our experiment. This is an interesting topic of further research which we hope can be built upon the proposed InImNets in the present article.
>
>
> 3. ''The analysis what depth of the network is required to learn the bounce off the walls in the bouncing ball experiment. Perhaps instead network interpretation, the authors could highlight and experiment with other benefits of the model: 1) we can take the output of the network after every and decode it into a meaningful output 2) we can do smooth interpolation between the input and the output.''
>
> Answer:
> While the discrete implementation with independent weights for each InImNet layer would not allow for smooth interpolation between layers as it is, we carried out the experiments with coupled weights (see Section 4.2 and Appendix F.3). In that setting, we demonstrate the ability of the model to decode outputs for intermediate layers and extrapolate them beyond optimised depths.
>
> In this version, we see competitive performance for rotating MNIST as well as the impact of going beyond p_min. We train the model on its outputs at the layer p=p_min=-4 and demonstrate the dynamics of total loss depending on the p in Figure 5. One can see that the accuracy decreases for higher layers.  It can also be observed that the extrapolation in higher layers (such as -8) is accompanied by decrease in sample diversity across the sequence.
>
> For the bouncing balls task (see section F.3 for the description), we see that coupling of the weights makes it more challenging to predict the balls' dynamics, however the model gives us the insight into what it learnt. We train the model on the outputs at the layers p=p_min=-3. In figure 7, it is shown that although having failed to model the dynamics, it converges to the strategy of erasing the sequence. For the extrapolated layers, it results in the reversal of the original sequence.

---

> > ### Comment · Reviewer_4Mn5 · 2021-11-22
> > **Good paper**
> >
> > Thank you for providing the additional experiments and further clarifications, particularly on the memory requirements. My score remains the same (8)

---

> ### Author Response · Authors · 2021-11-20
> **Response to referee (part 2)**
>
>
> 4. ''What happens if we go beyond p_max in the experiments?''
>
> Answer:
> We explore this in detail in section 4.2 and F.3.
>
>
> 5. ''Do you actually see high memory cost to justify using the finite difference method that is as accurate as the chosen delta? All experiments use MLP with a few layers?''
>
> Answer:
> Equation 21 outlines the approximation allowing to avoid high memory costs for training and inference of Jacobians. The aforementioned costs are associated with the recurrent computation of \nabla_{\xb} \zb(t, p_i , \nabla_{\xb}) which would require computation of i-th order gradients. While it may be still possible to use the exact solution for some of the smaller problems with the small p_min, we use the approximation from Equation 21 in all discrete architecture experiments.
>
> To further clarify on this question for the continuous architectures, we added explanation on computational costs in section F1, F.2 and section F.5.
>
>
> 6. ''Can we just vary the t_max rather than conditioning on p?''
>
> Answer:
> We believe this refers to the symmetry of p and q=t_max. Superficially, the purpose of varying p is to vary the location of the input x. Given that q is arbitrary, what is really the case is that we are varying the length q-p (or depth) of the learning problem. Rewriting the equations/notation, one could dress this up as varying q=t_max. After some deliberation, we preferred to vary p and fix q as a root point at which the output is measured.
>
>
> 7. ''How do you choose q?''
>
> Answer:
> Arbitrarily (see (6) above).
>
>
> 8. ''In the results section, what does p_max represent? Why do you vary p_max, even though the model integrates on the interval [p, q]. Should it be p_min?''
>
> Answer:
> Yes, it should be p_min. Thanks tremendously for this spot. We have changed the text accordingly throughout the experimental section. We state in the beginning of Section 4 that 'without loss of generality, we assume p to range between [p_min, 0], where p_min is a negative value representing 'depth' of the network.'

---

### Official Review · Reviewer_Ynkf · 2021-11-01

**Correctness:** 4
**Technical Novelty And Significance:** 3
**Empirical Novelty And Significance:** 3
**Recommendation:** 8
**Confidence:** 3

**Main Review:**

The paper aims to tackle the problem of modeling a continuous time dynamic system, using a category of neural networks. This idea is part of a novel field of study, starting with Neural ODEs (Chen et al, 2018). Thus, this paper studies a very interesting and novel problem, in a way that is in tune with prior work on the same topic.

Strengths:
+ The mathematical formulation of the proposed model is solid. The continuous time behavior of InImNets is adequately analyzed, and derivations for the forward and backward steps of the model are included.
+ The motivation behind the proposed model is also very interesting. Instead of solving a two-point boundary conditions problem (as in regular Neural ODEs), the authors solve two initial value problems for the same task. This allows the implementation of a simpler and more efficient scheme to train the network.
+ The formulation of the problem also leads to links with optimal control theory. This is an exciting direction which, while not extensively explored in the paper, can lead to interesting future works.
+ The experiments considered are adequate in order to provide a proof of concept for this architecture, when compared to prior work on the field.

Weaknesses:
-	I believe that there are parts of the paper that could be reformulated to become simpler and easier for the reader to understand. In particular:
  -	Section 2.1 should appear before the introduction of the section right above (since it describes the overall form of the optimization problem the model tries to solve).
  -	Equations 21 and 22 describe an alternative scheme which is not actually used, and I believe they may be omitted without hampering the flow of that section.
  -	Equations 16 and 17 are more closely related to Section 3.2 rather than the one they are placed right now.
-	The experimental section is not as clear as the rest of the paper. Firstly, Section 4.1 does not include any results, other than Figure 1 (I understand that the goal is primarily to provide a toy example as a motivation, but its placement makes the reader think that there are associated numerical results). Secondly, while Section 4 seems to imply that the model used was the discretized form of Algorithm 1, as described in Section 3, in the conclusion the authors state that they provide results for two models, a discrete and a continuous one. These two models should be better explained, both structurally and in which experiments they are used precisely.

Minor comments/Typos:
-	In the figures/tables in the experimental section, it would be nice to explicitly state the source of each architecture, in a way that matches the figure/caption (i.e. using the name of the architecture directly).
-	I believe the notation would be better if $z(t;p,x)$ for the InImNets was $z(p,x)$ (or something similar, given that while they do depend on the variable $t$, it is not considered to be the primary variable of the model.
-	Typos:
  -	Line after equation 2: “a external” -> “an external”.
  -	Page 3, overview of the paper: “InImnets” -> “InImNets”.
  -	Page 3, at the very end: “satisfies” -> “satisfy”.
  -	Page 5, after equation 13: “accessible optimize” -> “accessible to optimize”.
  -	Page 5, second to last paragraph before 3.1: missing word after “demonstrating high”.

Questions:
-	As mentioned above, I would be grateful if the authors could provide some additional explanations on the models used in their experiments.


**Summary Of The Paper:**

This paper proposes a class of networks called Invariant Imbedding Networks (InImNets). This type of networks aims to model an ODE as a pair of initial value problems, with the variable being the starting point of the problem. This allows for a more efficient implementation of a model of a dynamic system. The proposed model is analyzed both theoretically and experimentally.

**Summary Of The Review:**

Overall, this is a good paper which provides an exciting novel architecture, with good theory and some simple experiments to justify the quality of this architecture, with comparisons to related work. My current issues lie with the clarity of writing and the explanations of the models used in the experiments. Given that these issues can be easily resolved during the rebuttal process, I lean towards accepting this work, pending some extra explanation provided by the authors.

---

> ### Author Response · Authors · 2021-11-20
> **Response to referee**
>
> We thank you for you suggestions. We have itemised specific the points below and answer them directly.
>
> 1. ''Section 2.1 should appear before the introduction of the section right above (since it describes the overall form of the optimization problem the model tries to solve).''
>
> Answer:
> Thank you for this suggestion. We have followed this guidance and reordered the exposition as described. See the updated manuscript from Section 2.
>
>
> 2. ''Equations 21 and 22 describe an alternative scheme which is not actually used, and I believe they may be omitted without hampering the flow of that section.''
>
> Answer:
> Agreed. We have reworks and refined the section containing this equations at this suggestion.
>
>
> 3. ''Equations 16 and 17 are more closely related to Section 3.2 rather than the one they are placed right now.''
>
> Answer:
> Agreed again, this has been incorporated more naturally in the preceding section.
>
>
> 4. ''Section 4.1 does not include any results, other than Figure 1 (I understand that the goal is primarily to provide a toy example as a motivation, but its placement makes the reader think that there are associated numerical results).''
>
> Answer:
> Figure 1 was indeed included to build an analogy with network depth handled by an InImNet and a time series problem.
> We have experimented with this precise set-up. However, a fundamental difficulty is incurred: nested Jacobian calculation increases exponentially with depth. This is not such an issue for terminal loss optimisations as one can use shallower depths as prototypes. However, running cost problems necessarily propagate deep into the network to apply their loss values.
>
> We include an example experiment (Section F.2) which describes the maximum depth our RAM capacity can cope with. This is too limitted to train successful networks.
>
> Nevertheless, we give an explicit architecture that describes such time-series problems; see Section D. And we expand upon the pitfalls in such investigations; see Section F.1. This includes the problem of numerical gradient solutions (which don't incur the RAM costs) becoming unstable.
>
> We provide a Google Colab notebook to exhibit our experiment. This is an interesting topic of further research which we hope can be built upon the proposed InImNets in the present article.
>
>
> 5. ''While Section 4 seems to imply that the model used was the discretized form of Algorithm 1, as described in Section 3, in the conclusion the authors state that they provide results for two models, a discrete and a continuous one. These two models should be better explained, both structurally and in which experiments they are used precisely.''
>
> Answer:
> We have reworked the experimental part of the paper, to emphasise which architectures we use and what the experimental conditions are. In the beginning of the Section 3, we outline which sections describe the discrete and continious architectures.
>
> All the experimental results in Section 4, as well as F.3 and F.4, use the discrete architecture. We revised section 4 and Appendix G to emphasise the implementation details for the discrete architecture. To address the shortcomings of the model structure explanations, we have revised sections 3.1 and 3.2 explaining the discrete architecture used in the experiments. In appendix G, we included references to the discrete architecture description, and provide the explicit information about the loss function. In all experiments except section F.2 we optimise the decoded output values of the model at the layer $p_{\min}$ using backpropagation. The continuous architecture, alongside with its limitations, is described in Sections C, with the experiments with a simple prototype model F.1 and F.2.
>
>
> 6. ''In the figures/tables in the experimental section, it would be nice to explicitly state the source of each architecture, in a way that matches the figure/caption (i.e. using the name of the architecture directly).''
>
> Answer:
> We have modified the captions of the figures in the experimental section so that they reference the details of the architecture.
>
> 7. ''I believe the notation would be better if  z(t; p, x) = z(p, x) (or something similar, given that while they do depend on the t variable).''
>
> This is a good observation. We considered this change at the outset, but maintained the z(t; p, x) convention for the following reasons: (1) To remain in-keeping with the literature. (2) To explicate clarity throughout the derivation. (3) Whilst t is fixed, typically t=q, various equations depend on t from the outset, for example Eq. (5) [given as Eq. (2) for t=q] holds for each t in [p, q]. Eq. (25) utilises this description, alongside many equations in the appendix.
>
> We could introduce the notation z_t so that the network in the main is modelled as z_q(p, x). And we are indeed happy to make such a change, or other suggestions, after the referee's second consideration. We keep z(t; p, x) for now with additional clarification in the text.
>
> 8. All typos amended, thank you.

---

> > ### Comment · Reviewer_Ynkf · 2021-11-25
> > **Thank you!**
> >
> > Thank you very much for the detailed responses to my comments. I retain my suggestion to accept this paper, as noted in my main review.
> >
> > Regarding the notation $z(t;p,x)$, I understand the reasoning on why it should be kept, and I agree with the comments of the authors on this, so I'm fine with keeping the notation as is.

---

### Official Review · Reviewer_W69g · 2021-11-02

**Correctness:** 3
**Technical Novelty And Significance:** 3
**Empirical Novelty And Significance:** 3
**Recommendation:** 8
**Confidence:** 3

**Main Review:**

The strength of the paper is that it introduces an alternative view to the neural ODE, by reducing the non-linear, vector-valued optimal control problem to a system of forward-facing initial value problems. They introduce the invariant imbedding method from mathematics community to achieve this. They also show that the resulting algorithm can perform well compared to state-of-the-art methods.

There are two weaknesses of the paper. First, the presentation of the paper can be improved. I believe the paper should be written in a self-contained manner such that for the readers who have not seen the Neural ODE and optimal control view can still easily follow the paper, which is not the case in the current form. For example, the authors introduce and then start to derive the properties about the adjoint $\Lambda(p,x)$ without ever defining or explaining its role. Because of this, Theorem 2 is hard to follow without knowing the exact definition of $\Lambda(p,x)$. One possibility is to provide some sort of background review in the Appendix. Second, I think another weakness of the paper is that it does not seem to provide a rigorous treatment. Even though there are Theorem 1-3, I could not find any technical assumptions. It is weird to see Theorems stated without any assumptions on the model, and that makes the readers to wonder the rigor of the paper.

**Summary Of The Paper:**

In recent years, continuous depth neural networks, such as Neural ODEs have helped understanding of ResNet in terms of non-linear vector-valued optimal control problems. The authors of the paper show that it is possible to reduce the non-linear, vector-valued optimal control problem to a system of forward-facing initial value problems, providing an alternative perspective on the problem. They demonstrate that this approach can be used in creating discrete and continuous numerical DNN schemes.

**Summary Of The Review:**

(1) I suggest the authors to include a section discussing the background of Neural ODE, adjoint method etc. in the Appendix.

(2) Technical assumptions should be added, and the proofs should be made rigorous (if possible) throughout the paper. Otherwise, the paper stands as a non-rigorous treatment of the subject.

(3) On page 7, '3' and on page 9 'Invariant Imbedding'  are typos.

(4) In the caption of Table 1, the stop somehow appears in the next line.

---

> ### Author Response · Authors · 2021-11-20
> **Response to referee**
>
> We thank you for you suggestions. We have itemised specific the points below and answer them directly.
>
> 1. ''I suggest the authors to include a section discussing the background of Neural ODE, adjoint method etc. in the Appendix.'' (Specifically to clarify the significance of Theorem 2.)
>
> Answer:
> We agree that this will greatly improve readability and understanding. Action we have taken: (1) extended paragraph 3; added paragraphs 4, 5; elaborated at item 2 in the introduction; and further reference in Section 2.3 to better explain the role of the adjoint and the connnection with the typical `optimisation-by-backpropagation'. We also note that this is succinctly described by Chen et. al. 2018 and so we refer to their proof. (2) We have included a section in the appendix to discuss the significance of the adjoint method from several angles. We had detailed the derivation via the method of Euler--Lagrange; we also now reference this in our new summary and also discuss/reference the proof given by Chen et. al. and application by Vialard et. al.
>
>
> 2. ''Technical assumptions should be added, and the proofs should be made rigorous (if possible) throughout the paper. Otherwise, the paper stands as a non-rigorous treatment of the subject.''
>
> Answer:
> We strongly agree. Beforehand we had reserved technical details to the appendix. We have brought all necessary assumptions forward to Section 2. These may be found in the first paragraph of Section 2 and then in the statement of Theorems 1, 2, 3. After Theorem 1, we also give discussion on possible relaxations of our assumptions and refer the literature for where their explicit statements are defined. These are now self contained mathematical assertions. The proofs of these three theorems are explicated in the appendix. This was previously a single linear derivation. We have amendments the sectioning/exposition to identify exactly how each proof is concluded.
>
>
> 3. Thank you for pointing out these typos in the first manuscript. All amended.

---

> > ### Comment · Reviewer_W69g · 2021-11-29
> > **response to authors response**
> >
> > Thank you for the response. I have raised your score from 6 to 8.

---

### Official Review · Reviewer_nSYs · 2021-11-03

**Correctness:** 3
**Technical Novelty And Significance:** 3
**Empirical Novelty And Significance:** 3
**Recommendation:** 6
**Confidence:** 4

**Main Review:**

This paper studies a very important and interesting topic, continuous depth neural networks. It may be one way to reach the framework of Explainable AI. I appreciate the authors providing a very detailed derivation and implementation for the InImNet architecture, which enables readers to easily reproduce the proposed method.

However, I still have some concerns, which are listed as below:

1. According to the current experiment results, I am not sure what is the superiority of the proposed InImNet compared to existing approaches. For example, in Table 1, both the InImNet and other existing models can achieve similar performance. The authors may want to give a detailed explanation about such superiority.

2. For the "rotating MNIST" experiments, only the MLP model is used for implementing InImNet. I would be happy to see InImNet being implemented with other types of models, for example, a small CNN. Besides, can the authors apply InImNet to some large-scale datasets like CIFAR-10?

3. According to the implementation of InImNet, it seems would take more calculation consumption for training the InImNet. Therefore, it is necessary to compare the time used for training InImNet and other models. Further, I would like to see the comparison of performance for different models under the same training time cost.

**Summary Of The Paper:**

This paper proposes a new type of continuous depth neural network architecture, named Invariant Imbedding Networks (InImNets). InImNets can imbed a given $\bf x$ input to different time steps (time positions) and efficiently calculate the solutions of the dynamic system (starting from different time positions) simultaneously. The authors also give a practical implementation for training the InImNets architecture. The proposed architecture is then applied to several benchmark problems for continuous neural networks for verifying its effectiveness.

**Summary Of The Review:**

Based on the previous comments, I tend to recommend accepting this paper. But I would like the authors to resolve my concerns first.

---

> ### Author Response · Authors · 2021-11-20
> **Response to referee**
>
> We thank you for you suggestions. We have itemised specific points below and answer them directly.
>
> 1. ''What is the superiority of the proposed InImNet compared to existing approaches? For example, in Table 1, both the InImNet and other existing models can achieve similar performance. The authors may want to give a detailed explanation about such superiority.''
>
> Answer:
> We agree with the reviewer's suggestion to better explain the advantages of InImNet. We have expanded on the previous empirical analysis, measuring the time per epoch for the experiments for the discrete model: in section 4.1 of the revised paper, we demonstrate the improvements in computational efficiency of the discrete model over the state-of-the-art models. Also, in sections 4.2 and F.3 as analyse the ability of the InImNet architecture to produce interpretable intermediate outputs and extrapolate beyond the trained 'depth'; in Section F.4 we demonstrate the ability of the architecture to be composed of multiple convolutional layers and produce meaningful intermediate outputs. In addition, as we outline in the answer to the next question and in Section F.4, we implemented a small CNN model for the bouncing balls experiment, which shows ability of the discrete architecture to cope with more high dimensional data than it was reported in the state-of-the-art papers such as Vialard et al (2020).
>
> 2. ''For the 'rotating MNIST' experiments, only the MLP model is used for implementing InImNet. I would be happy to see InImNet being implemented with other types of models, for example, a small CNN. Besides, can the authors apply InImNet to some large-scale datasets like CIFAR-10?''
>
> Answer:
> We think that adding analysis for other types of models, for example a small CNN, is a really good suggestion. We implemented it and provide the details in the Appendix E.2 and a summary here.
>
> We implemented a fully convolutional autoencoder structure for the bouncing balls task, replicating the setting of the bouncing balls experiment in the first version of the paper, however we incorporate these autoencoders as a part of the InImNet model. This results in a bigger model, which operates fully convolutionally and therefore does not need preprocessing to map the data into the latent space. We incorporate the time variable as a separate channel of the autoencoder input. The results show improvement over the multilayer perceptron results (MSE of 0.0122 against 0.0192 for the InImNet).
>
> Also, we agree that CIFAR-10 is a suitable benchmark dataset. Due to the time constraints of the submission, we have focused on expanding other areas where possible. We shall nevertheless continue to apply our InImNets to larger datasets such as CIFAR-10 to add to our GitHub repo on ongoing research.
>
> 3. ''According to the implementation of InImNet, it seems would take more calculation consumption for training the InImNet. Therefore, it is necessary to compare the time used for training InImNet and other models. Further, I would like to see the comparison of performance for different models under the same training time cost.''
>
> Answer:
> Large calculation memory consumption is indeed an obstacle for the continuous implementations of InImNets. We have now included experiments in the submission that quantify such observations (see section F.2 and F.5 for the details). The large processing demands for the continuous architectures are a cost which we plan to refine in future work (for instance by numerical gradient approximation). However, this is a cost that purchases a significant theoretical advantage, in that each output models a network of different input location. So a reasonable comparison would be to compare an InImNet's consumption with a collection of stand-alone fixed-input networks, one for each layer in the InImNet. We consider this point of view in the updated article.
>
> For the discrete implementation of InImNet, we expanded on time analysis given in the first version, and we show below and in sections 4.1.2 and 4.1.3 the time consumption for different number of layers; we show that we can achieve competitive results in a more time efficient way than the state-of-the-art continuous model Vialard (2020).
>
> The results for average time per epoch are included for comparison in the benchmarks of Section 4.1.

---

> > ### Comment · Reviewer_nSYs · 2021-11-29
> > **Thanks for response and revision**
> >
> > I appreciate the detailed response and revision from the authors. I find this paper is of high significance in novelty and theoretical analysis but has relatively limited advantages in the empirical aspect. I recommend accepting this paper and keep my score 6.

---

### Author Response · Authors · 2021-11-20
**Response to referees: overview of amendments**

We thank the referees for their time and careful consideration of our article. We have incorporated the proposed suggestions into the updated text, providing additional details and refinements. We feel the quality of the paper has been substantially improved by the referee's feedback, which has lead to additional, and extensive, experimental analysis and a reworking of the theoretical presentation.

We detail specific responses to individual referees below. Here, we give a general summary of the amendments made in the updated text.


1. Additional experimentation: A common remark was that the experimentation section was limited. We have extended this as far as possible within the response time. The updates are predominantly found in Sections 4, F and G. More specifically, we focussed on:

- The impact of extrapolation below and beyond depths at which we optimise the network ("what happens if we go beyon p_min?"); see Sections 4.2 and appendix section F.3.

- The use of convolutional architectures in the dynamics functions for learning problems; see Section F.4.

- Enhanced performance comparisons in existing benchmark runs, for example time per epoch stats.

- An assessment of InImNets applied to time-series problems where the underlying ODE is learnt directly by the training function ODE; see Section F.2.

- Further specifics on the hyperparameters and experimental set-up; see Section G and further comments in situ.


2. Expository amendments:

- We have reworked Section 2 to better describe the theoretical set-up.

- We have explicitly stated the hypotheses necessary for each theorem to hold, as well as further discussion on hypotheses requirements (including reference to where this is further explained).

- We have included section D.1 in the appendix to give a high-level background on the adjoint method. We also included highlights of this in the introduction, particularly in the run-up to Theorem 2 where the main theorem on the adjoint is stated.

- We have amended all indicated typos, and a further few bonus ones.

---

### Public Comment · ~Dmitry_Kangin1 · 2022-02-14
**Code**

We release the accompanying code for the paper in the following repository:

https://github.com/andrw3000/inimnet

---

### Decision · Program_Chairs · 2022-01-20

**Decision:**

Accept (Spotlight)

**Comment:**

The authors presents an alternative view of Neural ODEs, offering a novel understanding of depth in neural networks. The reviewers were overall impressed by the novelty and potential for insights this work brings. There was some disappointment that the empirical results were not stronger (both in terms of pure performance and computational cost) and that it wasn't clear how the theoretical insights into depth actually translated into a practical insight. Nevertheless, I agree with the reviewers that this is a good submission and would I think make for an interesting addition to the conference programme.